**Automated urban flood level detection based on flooded bus dataset using YOLOv8**
Yanbin Qiu[a], Xudong Zhou[b], Jiaquan Wan[a,*], Tao Yang[a,c], Lvfei Zhang[a], Yuanzhuo Zhong[a], Leqi Shen[a],
Xinwu Ji[a]
[a] College of Hydrology and Water Resources, Hohai University, Nanjing 210024, China
[b] Institute of Hydraulics and Ocean Engineering, Ningbo University, Ningbo 315000, China
[c] Institute of Water Resources and Technology, Hohai University, Nanjing 210024, China
**Corresponding Author:**
Jiaquan Wan
College of Hydrology and Water Resources, Hohai University, Nanjing 210024, China
Email: 775290095@qq.com
**ABSTRACT**
Rapid and accurate acquisition of urban flood information is crucial for flood prevention, disaster
mitigation, and emergency management. With the development of mobile internet, crowdsourced images on
social media have emerged as a novel and effective data source for flood information collection. However,
selecting appropriate targets and employing suitable methods to determine flooding level has not been well
investigated. You Only Look Once version 8 (YOLOv8) is a convolutional neural network-based computer
vision model that has been widely applied in image recognition tasks due to its end-to-end architecture and
high computational efficiency. This study proposes a method to assess urban flood risk levels based on the
submerged status of buses captured in social media images. First, a dataset containing 1008 images in
complex scenes is constructed from social media. The images are annotated using Labelimg, and expanded
with a data augmentation strategy. Four YOLOv8 configurations are validated for their ability to identify
urban flood risk levels. The validation process involves training the models on original datasets, augmented
datasets, and datasets representing complex scenes. Results demonstrate that, compared to traditional
reference objects (e.g., cars), buses exhibit greater stability and higher accuracy in identification of urban
flood risk levels due to their standardized height and widespread presence as they remain in service during
flood events. The data augmentation strategy enhances the model's mAP50 and mAP50-95 metrics by over
10% and 20%, respectively. Additionally, through comparative analysis of YOLOv8 configurations,
YOLOv8s demonstrates superior results and achieves an effective balance between accuracy, training time,
and computational resources, recommended for the identification of urban flood risk levels. This method
provides a reliable technical foundation for real-time flood risk assessment and emergency management of
urban transportation systems, with substantial potential for practical applications.

**Keyword: Urban Flooding; Automatic Detection; Computer Vision; Object Detection**

**1. Introduction**

With the intensification of global climate change, extreme precipitation events have increasingly triggered urban pluvial flooding, severely disrupting the operation of major cities (Guan et al., 2015). Concurrently, the proportion of impervious surfaces has been rising due to rapid urbanization, significantly diminishing the infiltration capacity of urban landscapes (Cao et al., 2025; Zhengzheng et al., 2025) and resulting in an increase in surface runoff (Chaudhary et al., 2020). This has led to more frequent urban flooding incidents, imposing substantial impacts and losses on urban infrastructure, transportation networks, and human wellbeing. For instance, on July 20, 2021, a rare extreme rainfall event, with a record-breaking maximum hourly rainfall of 201.9 mm, driven by Typhoon In-Fa struck Zhengzhou, Henan Province, China, leading to severe urban inundation that resulted in 292 casualties and direct economic losses reaching 53.2 billion (Yang and Wang, 2022).

In the event of urban flooding, the ability to rapidly and accurately identify flood risk levels is crucial for urban flood prevention, mitigation, and emergency response decision making (Fohringer et al., 2015; Qian et al., 2022; Smith et al., 2017). Flood depth is widely regarded as the most representative indicator for assessing the severity of flood impacts. (Betterle and Salamon, 2024). The prediction of flood depth helps guide evacuation planning and resource allocation, providing decision support for emergency response and thereby significantly reducing loss of life and property damage (Jiang et al., 2020; Park et al., 2021). (Alizadeh Kharazi and Behzadan, 2021) pointed out that even 2.54 cm of floodwater can cause approximately $27,000 in combined damages to an average one-story home. When flood depth ranges between 30 to 60 cm, the risks of vehicle flotation and stalling increase substantially (Kramer et al., 2016; Pregnolato et al., 2017; Wang et al., 2025), posing serious threats to human safety. Moreover, flood depth information on various roads during rescue operations is essential as it influences resource allocation and rescue route planning. For example, during Hurricane Katrina, emergency responders often requested information regarding the flood extent and water depth in order to deploy appropriate vehicles for search and rescue (SAR) operations and to identify the optimal routes to reach victims (Nayak and Zlatanova, 2008). For example, in Texas during the Hurricane Katrina flooding event, it was estimated that approximately 75% of flood-related fatalities occurred during evacuation efforts via local roadways, primarily due to the lack of awareness regarding inundation depth in the surrounding areas. (Alizadeh Kharazi and Behzadan, 2021).

Consequently, flood depth prediction plays a vital role in assessing road passability, delineating road closures,
optimizing rescue logistics, and prioritizing areas for emergency intervention (Jiang et al., 2020; Kundu et
al., 2022).

64         Currently, urban waterlogging monitoring primarily relies on water level gauges (Fohringer et al., 2015).

Although water level gauges can monitor flood depth in real-time, their deployment and maintenance are
costly, and the monitoring range is restricted by the installation locations, limiting their suitability for wider
spatial coverage (Chaudhary et al., 2020; Fohringer et al., 2015; Paul et al., 2020). Microwave remote sensing
methods have limitations in spatial-temporal resolution and data frequency, and are susceptible to
interference from clouds and obstructions, rendering them unable to determine flood depth (Chaudhary et
al., 2020; DeVries, 2020; Liang, 2020). An intelligent and low-cost technology capable of identifying urban
flood risks with extensive spatial coverage is urgently needed.

72         In recent years, with the rapid development of social media and mobile internet, the application of social

media data in flood monitoring and risk assessment has garnered extensive attention (Baranowski et al., 2020;
Kankanamge et al., 2020; Li et al., 2023; Rosser et al., 2017; Smith et al., 2017). Platforms like Weibo,
Twitter and Douyin provide users with channels to share flood information in real time, where user-generated
content (UGC) contains rich flood imagery and geolocation data, offering a novel data source for urban flood
level detection research (Iqbal et al., 2021). Concurrently, significant advancements have been made in
computer vision technology, particularly in the application of convolutional neural networks (Voulodimos
et al., 2018) , opening new avenues for the analysis of vast amounts of flood imagery data. Current studies
have attempted to use objects in social media images, such as bridges (Bhola et al., 2018), roadside barriers
(Jiang et al., 2019), bicycles (Chaudhary et al., 2020), traffic cones (Jiang et al., 2020), traffic signs (Alizadeh
Kharazi and Behzadan, 2021), water level markers (Jafari et al., 2021), and pedestrians (Li et al., 2023), as
reference points for flood level estimation. While specific reference objects have shown promising results
in studies, their infrequent occurrence hinders their broad application in urban settings. Pedestrians are
prevalent in urban areas, but their low image resolution and diminished presence in severely flooded zones
reduce their reliability and practicality as reference points for water level estimation.

87         Vehicles serve as ideal reference objects for recognizing urban flood levels, attributed to their stable

morphological features, widespread availability, and ease of detection. Current research leverages vehicles
for urban flood water level identification. For example, (Park et al., 2021) and (Huang et al., 2020) used
Mask R-CNN to detect the submerged state of vehicles or their wheels as an indicator of flood levels. Wan
et al. utilized the YOLO (You Only Look Once) series models, a CNN-based computer vision (CV) model,
for urban flood risk assessment and detection (Puliti and Astrup, 2022; Redmon et al., 2016; Wan et al.,
2024; Zhong et al., 2024). However, most studies use cars as reference objects; yet, the diversity of car types
(e.g., sedans, SUVs, and pickup trucks) introduces significant variations in height and dimensions, affecting
model generalization. Furthermore, the limited height of car bodies means they cannot provide effective
water level information once submerged up to the roof, and their lower frequency of appearance in extreme
weather makes it challenging to collect image datasets.

In comparison, buses, as a critical component of urban transportation systems, possess standardized

heights and structures with minimal variation between models, making them a more ideal reference object
for flood water level monitoring. Buses overcome the limitations posed by cars, such as variations in size
and limited height. Additionally, buses primarily operate in busy or essential areas, and their ability to
withstand submersion is crucial for the continued operation of the urban public transportation system. Flood
level recognition based on the submerged status of buses can intelligently assess their water-related risks,
providing valuable support for urban transportation emergency management.

In response to the gaps in existing research, this study aims to create a comprehensive dataset of

submerged buses by sourcing flood images from social media platforms. Based on urban flood safety
standards and bus height characteristics, the submerged states of buses are categorized into specific levels.
The dataset includes complex scenes (e.g., nighttime, occlusions, and incomplete bus bodies) to enhance
data diversity.

YOLO is an efficient real-time object detection algorithm that simplifies the traditional multi-stage

detection pipeline by framing object detection as a regression problem. In the YOLO model, the image is
divided into a grid, with each grid cell predicting the location (bounding box) and class probability of an
object. The uniqueness of YOLO lies in its ability to predict multiple bounding boxes simultaneously in a
single forward pass, significantly improving computational efficiency and processing speed. The model
utilizes a convolutional neural network for feature extraction and considers the global context of the image,
which helps reduce the likelihood of false positives. The YOLOv8 algorithm, as the most recent iteration of
the YOLO series, has exhibited outstanding accuracy and rapid detection performance on the standard
COCO (Common Objects in Context) dataset (Wan et al., 2024). In this study, the YOLOv8 model is trained
on the purpose-built dataset to improve its performance in identifying urban flood water levels with precision.
This study introduces an innovative approach to flood level detection by leveraging the submerged
states of buses, addressing the limitations of traditional recognition methods based on cars and other
reference objects and overcoming the limitations of conventional monitoring techniques in broader
applications. Specifically, the objectives of this study include:
1)   Developing a comprehensive dataset of submerged buses to examine the relationship between bus

submersion and flood water levels.

2)   Evaluating the performance of YOLOv8 configurations in assessing flood severity through original and

augmented data training, as well as experiments involving complex scenes.

3)   Proposing configuration recommendations for YOLOv8 aims to address diverse application scenarios,

ensuring efficient deployment in varied urban environments.

This paper is structured as follows: Section 2 provides a detailed description of the dataset construction,
data augmentation strategies, YOLOv8 model configurations and explains the experimental design and
model evaluation metrics. Section 3 explains experimental results, followed by a comparative analysis of the
findings. Section 4 discusses the experimental results and offers configuration recommendations for
YOLOv8. Finally, Section 5 provides a summary of the main conclusions and highlights potential directions
for future research.
**2. Methodology**
**2.1 Data acquisition and processing**
**2.1.1 Data acquisition**
In this study, a comprehensive dataset of submerged buses was constructed, comprising 1,008 images
that capture buses in various statues of submersion. These submerged bus images were collected through
keyword searches such as "urban flooding" and "submerged bus" on Baidu and Google, screenshots of
relevant frames in short videos on Douyin, browsing urban flood news, and obtaining images from WeChat
public accounts. The data collection generally focused on images from the past five years. Due to the diverse
sources of image data, the images used in the experiment vary in resolution and size.
All collected images were manually screened based on the following criteria: (1) the bus features (e.g.,
license plate, doors, wheels) must be clearly discernible to the human eye; (2) images without a bus were
excluded; (3) grayscale or monochrome images were excluded; (4) buses were either stationary or normally
moving through floodwater during flooding events, without signs of floating or water-induced displacement.
These screening rules ensured the validity and reliability of buses as stable reference objects.
On this basis, when selecting bus images, this study included complex and challenging scenes, such as
nighttime scenes, partial occlusions, and incomplete buses, to enhance the dataset's representativeness. The
inclusion of these scenes may help enhance the diversity of the dataset, potentially enabling the trained model
to better handle various flood scenes. Table 1 presents the number and proportion of images collected from
each platform.
**Table 1**
Distribution of collected images by platform

| Platform | Number of images | Proportion (%) | Time range |
|----------|------------------|----------------|------------|
| Baidu | 260 | 26 | 2020-2024 |
| Google | 437 | 43 | 2020-2024 |
| Douyin | 218 | 22 | 2020-2024 |
| Others | 93 | 9 | 2020-2024 |

**2.1.2 Data annotation**
This study assesses flood risk levels based on the submersion status of buses. In this study, the
approximate height of buses was uniformly set to 3 meters in the model, based on the specifications of
commonly observed urban bus types in social media images. Based on a systematic evaluation of the images,
it was found that the submersion depths of buses were mostly concentrated between 0 and 50 cm, with values
exceeding 100 cm observed only in a few cases. Given that high water levels (greater than 100 cm) are
relatively rare in urban scenarios, this study chooses to define classification levels within the more commonly
observed water depth range. Consequently, four flood levels were established, with detailed information on
each flood level and corresponding bus examples provided in Table 2. Using Labelimg, a widely used
annotation tool for object detection model training, the 1,008 acquired images were annotated to identify
1,562 bus instances, which were stored in YOLO format for subsequent training. Notably, the number of
Level4 instances is comparable to that of the other levels (see Table 2). This is because the few images in
our dataset with water depths exceeding 100 cm contained a relatively large number of Level4 bus-
submersion instances. Based on different states of bus movement, annotation types were divided into two
categories: holistic and segmented. In holistic annotation, the entire bus is assigned a single flood level,
while in segmented annotation, specific parts of the bus correspond to different levels, as illustrated in Fig.

1.

**Table 2**
Flooded bus dataset: bus instances and flood levels.

| Flood levels | Analysis of bus submersion depths | Range of water depth | Number of instances | | |
|---|---|---|---|---|---|
| | | | total | validation set (n=108) | validation set (n=198) |
| Level1 | bottom of wheels submerged; inner wheel contour not visible | 0~20 cm | 296 | 42 | 67 |
| Level2 | between bottom of wheels and halfway up the tires, between bottom of wheels and top edge of license plate, water reaches 1/4 of the step at bottom of door | 20~45 cm | 585 | 48 | 129 |
| Level3 | between halfway up the tires and full tire coverage, between top edge of license plate and bottom edge of windshield | 45~100 cm | 371 | 43 | 72 |
| Level4 | above the bottom edge of the windshield, fully submerged tires | >100 cm | 310 | 35 | 61 |

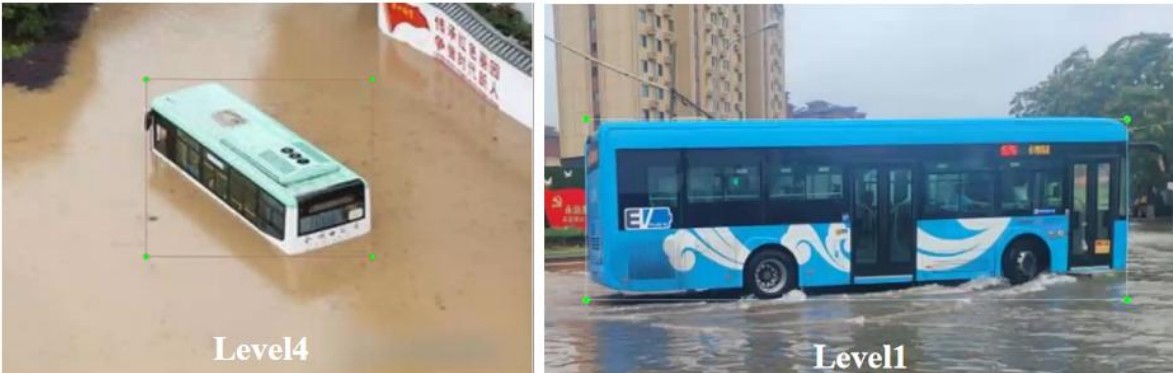

(a) The entire bus is assigned a single flood level

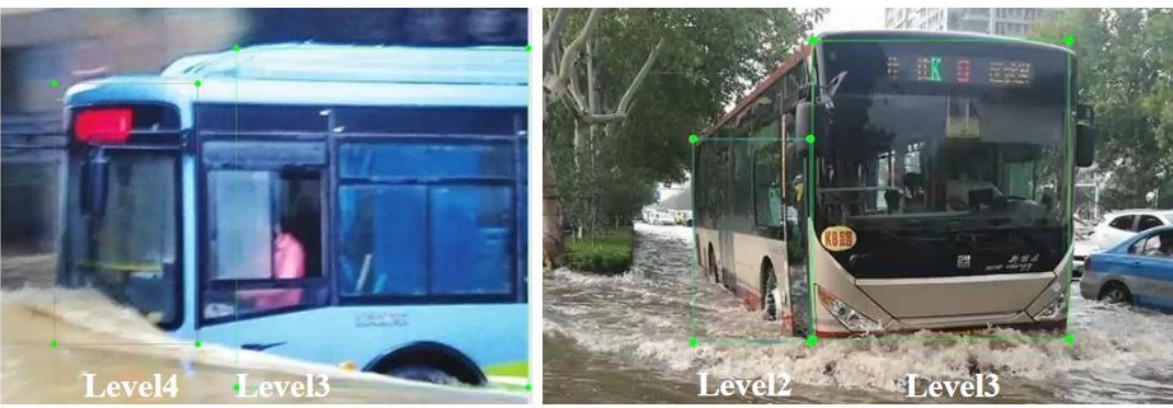

(b) The specific part of bus is assigned different levels

**Fig. 1.** Examples of dataset annotations: (a) The entire bus is assigned a single flood level; (b) The specific part of bus is assigned different levels.

### 2.1.3 Data augmentation

The original dataset, comprising 1,008 images, is relatively small in scale, necessitating its

enhancement and expansion to increase background complexity, prevent overfitting, and improve robustness.

Image data augmentation methods can be broadly categorized into two types: luminosity distortions and

geometric distortions(Li et al., 2023). The former involves adjusting image brightness, contrast, hue,
saturation, and adding noise, while the latter encompasses random scaling, cropping, flipping, and rotation
operations. In this study, horizontal and vertical flipping, rotation, and random cropping were applied to
augment the original dataset, as illustrated in Fig. 2. All images in the augmented dataset were annotated
using Labelimg and stored in YOLO format. Additionally, considering the potential for manual annotation
bias, a review process was conducted to verify the correctness of all bounding boxes and annotations related
to flood risk levels.

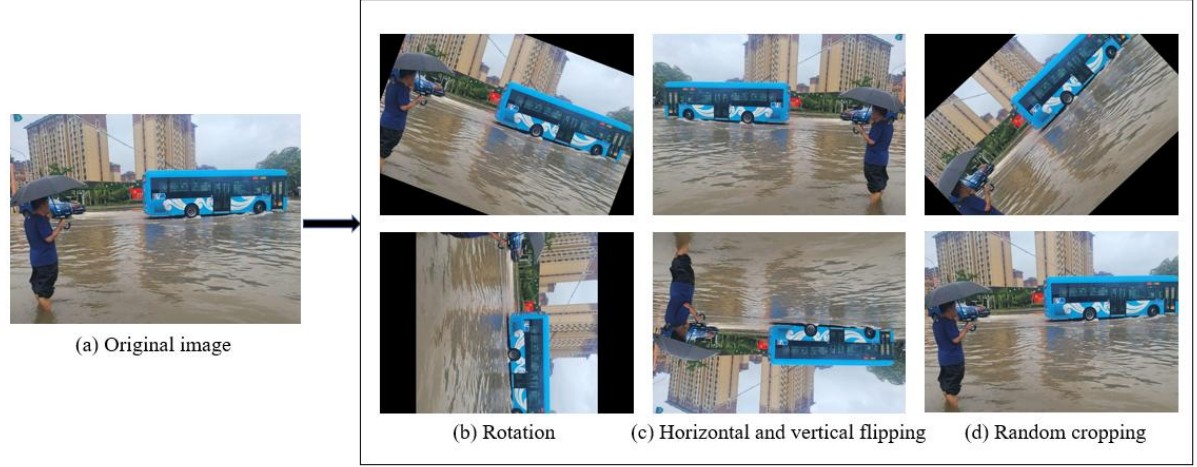

**Fig. 2.** Examples of dataset augmentations: (a) Original image; (b) Rotation; (c) Horizontal and vertical
flipping; (d) Random cropping.
**2.2 Object detection model**
A key feature of the YOLO model is its ability to achieve an optimal trade-off between speed and
accuracy, enabling rapid and precise object detection across a wide range of application scenes (Wan et al.,
2024). This study constructed an object detection model based on the YOLOv8 source code, which operates
on convolutional neural networks (CNN). YOLOv8 is an integrative and enhanced version building on
previous YOLO generations and represents the latest iteration in the YOLO series. This version significantly
improves computational efficiency and inference speed by optimizing the network architecture and refining
inference algorithms. Moreover, YOLOv8 demonstrates higher stability in multi-object detection tasks under
complex scenes, particularly with its advanced features such as automated hyperparameter tuning and
dynamic convolution modules. These enhancements further boost the model's flexibility and adaptability,
making it more capable of meeting the diverse requirements of real-world applications.
The YOLOv8 processing workflow includes image preprocessing, multi-level feature extraction
through CNN, and multi-scale feature fusion via a feature pyramid and path aggregation network. Following
this, adaptive anchor boxes are used for bounding box regression and classification prediction, with non-
maximum suppression applied to eliminate redundant bounding boxes. The model ultimately outputs the
target's class, bounding box, and confidence score, ensuring a balance between detection accuracy and
efficiency.
As shown in Fig. 3, the YOLOv8 network architecture is composed of three primary parts: Backbone,
Neck, and Head. The Backbone consists of five convolutional modules, four C2f modules, and a Spatial
Pyramid Pooling-Fast (SPPF) module, all designed for feature extraction. The Neck refines and integrates
features derived from the Backbone, improving both the precision and reliability of object detection. The
Head utilizes a decoupled structure to handle feature maps across multiple scales, generating the final
detection outputs.

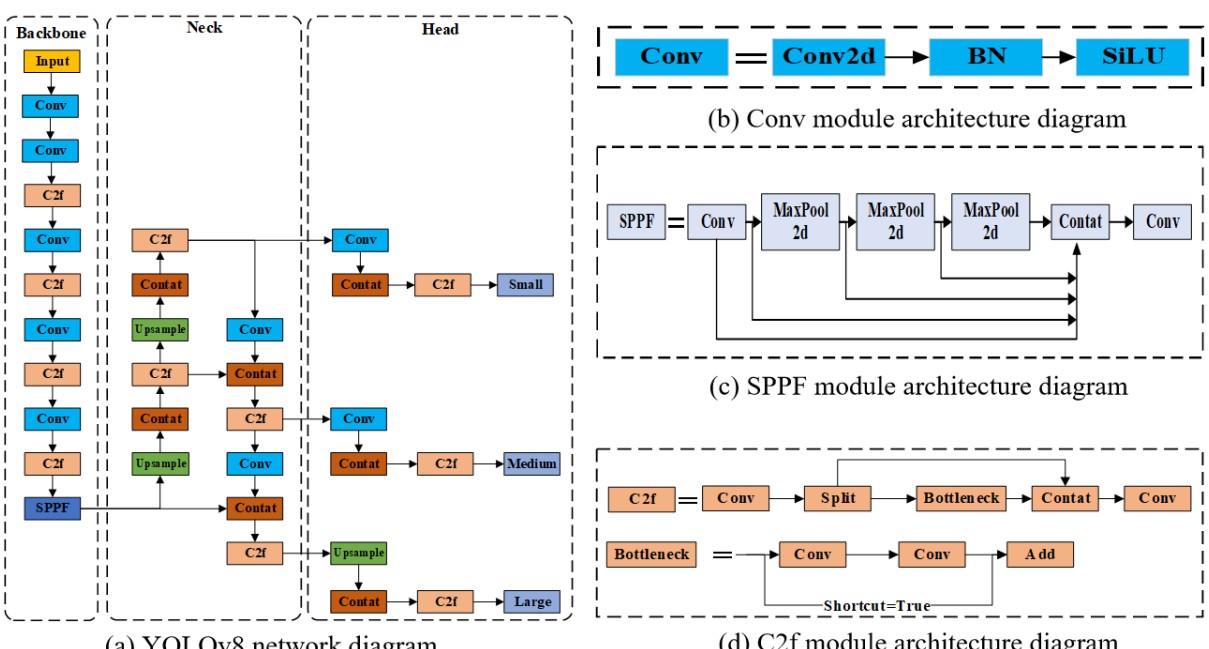

(a) YOLOv8 network diagram        (d) C2f module architecture diagram


**Fig. 3.** The YOLOv8 network structure: (a) YOLOv8 network diagram; (b) Conv module architecture
diagram; (c) SPPF module architecture diagram; (d) C2f module architecture diagram.
**2.2.1 Experiment on object detection model**
Following data augmentation, the original dataset was expanded to 2,184 images for model training. In
this study, four different YOLOv8 configurations—YOLOv8n, YOLOv8s, YOLOv8m, and YOLOv8l—
were used for training, each designed with specific network depths and widths to address varying application
requirements. Table 3 presents the key characteristics of these four YOLOv8 configurations. The number of
layers reflects the depth of the model, with a greater number of layers indicating a deeper model capable of
capturing higher-level semantic features. However, this also leads to increased computational complexity.
Parameters refer to the total number of learnable weights and biases within the model. A higher number of
parameters signifies stronger representational capacity, making the model suitable for more complex tasks,
but it also requires greater computational resources and longer training times.
**Table 3**
The primary characteristics of the four YOLOv8 configurations.

| Model | Size | Layers | Params (M) | mAP50-95 on val (%) | FLOPs (B) | Speed CPU ONNX (ms) | Speed A100 TensorRT (ms) |
|---|---|---|---|---|---|---|---|
| YOLOv8n | 640 | 225 | 3.2 | 37.3 | 8.7 | 80.4 | 0.99 |
| YOLOv8s | 640 | 225 | 11.2 | 44.9 | 28.6 | 128.4 | 1.20 |
| YOLOv8m | 640 | 295 | 25.9 | 50.2 | 78.9 | 234.7 | 1.83 |
| YOLOv8l | 640 | 365 | 43.7 | 52.9 | 165.2 | 375.2 | 2.39 |

**2.2.2 Experimental setup**
The model was implemented using PyTorch, a framework offering libraries for object detection models.
Training was performed on NVIDIA Quadro RTX 3090 GPU. The original dataset contained 1,008 images,
with 90% allocated for training and the remaining 10% for validation. The initial input size was set to $640 \times$
640, with a batch size of 16 and 100 training epochs. In the data augmentation experiment, augmentation
was applied only to the training set, resulting in 1,986 augmented training images. To maintain a consistent
90%-10% split ratio, the validation set comprised 198 images, while all other settings remained the same as
previously described. The distribution of the validation set across the 4 levels is summarized in Table 2.
Throughout the entire training process, the validation set was strictly separated from the training set, and the
model had no access to any original or augmented images from the validation set. In this study, the official
default hyperparameter settings of YOLOv8 were adopted (Sun et al., 2023; Yu et al., 2025). These values
are widely validated by the Ultralytics team across multiple benchmark datasets and tasks, providing a stable
balance between accuracy, convergence speed, and computational efficiency, and are designed to be
hardware-friendly for common GPU configurations.
**2.2.3 Complex scenes prediction experiment**
The performance of object detection models on the validation dataset does not fully represent its overall
capabilities, as view of the bus, impacted by camera angle and distance, introduces numerous sources of
interference. To address this, a complex scene prediction experiment was designed in this study to assess the
detection capabilities of the different YOLOv8 configurations under challenging urban flood environments.
In this experiment, YOLOv8 models trained on either the original or augmented datasets were used to
conduct complex scene predictions, evaluating their performance under challenging conditions. Currently,
there is no publicly available large-scale dataset of bus flood inundation images, and the images obtainable

from social media predominantly depict routine scenes, whereas extreme, complex scenes are rare and dispersed, making it infeasible to construct an independent large-scale evaluation dataset. Therefore, this study selected two particularly demanding scenes for experimental evaluation. Fig. 4 presents two particularly demanding scenes, not in the existing dataset, Fig. 4a shows a rainy scene with multiple vehicles, where the incomplete view of the bus, impacted by camera angle and distance, introduces numerous sources of interference, increasing detection complexity. Fig. 4b depicts a nighttime scene of a submerged bus, where low light and poor image quality significantly elevate the difficulty of detection.

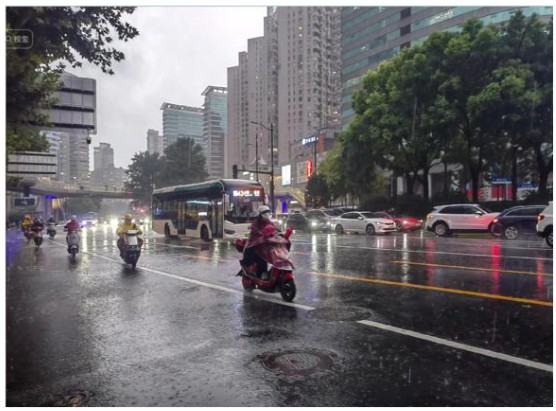 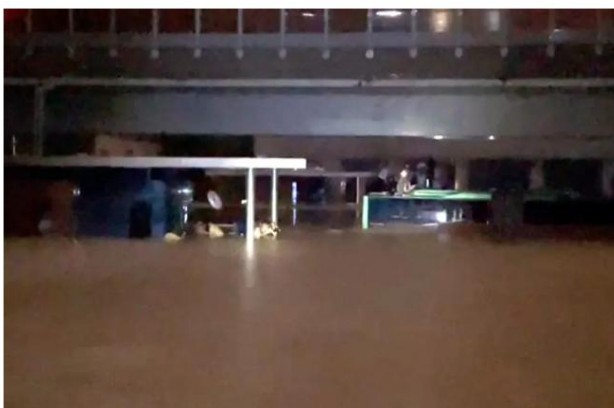

(a) Low flood risk scene with multiple vehicles present     (b) High flood risk scene with blurring and corruption

**Fig. 4.** Two complex urban flooding scenes: (a) Low flood risk scene with multiple vehicles present; (b) High flood risk scene with blurring and corruption.

### 2.2.4 Comparative experiment with YOLOv5 model

Although the introduction states that YOLOv8 is the latest algorithm in the YOLO series and has been known to perform better than earlier versions on a general image dataset, comparative analysis with earlier versions was performed for this dataset to quantify performance differences.This study will compare and analyze the configuration demonstrating superior performance in the experiments with the corresponding configuration in the YOLOv5 model. The experiments include training on the original dataset, training with data augmentation, and complex scene prediction, following the same experimental setup described earlier.

### 2.3 Model evaluation

The widely recognized evaluation metrics, such as precision (P), recall (R), and F1 scores are as follows:

$$Precision = \frac{TP}{TP + FP} \tag{1}$$

$$Recall = \frac{TP}{TP + FN} \tag{2}$$

$$F1 = \frac{2 \times Precision \times Recall}{Precision + Recall} \tag{3}$$

Precision (P) quantifies the model's false detection rate, while Recall (R) evaluates its true detection rate. The F1-score evaluates model performance, particularly in object detection tasks, as it combines both P and R, representing their harmonic mean, where True Positive (TP) refers to the count of objects correctly identified as positive, True Negative (TN) denotes the count of non-objects accurately classified as negative, False Positive (FP) indicates the count of non-objects incorrectly classified as positive, and False Negative (FN) represents the count of objects wrongly classified as negative.

In this study, mean Average Precision (mAP), a well-established evaluation indicator, is utilized as the primary criterion to evaluate the performance of different YOLOv8 configurations in detecting bus submersion states. The mAP is computed by summing the average precision across all labels and dividing the result by the total number of categories. A higher mAP value signifies improved average accuracy of the model, indicating enhanced overall detection performance. The formula for calculating mAP is as follows:

$$mAP = \frac{1}{n}\sum_{i=1}^{n} AP_i = \frac{1}{n}\sum_{i=1}^{n}\int_{0}^{1} P_i(R_i)dR_i \tag{4}$$

where n is the number of categories, Average Precision (AP) is the area under the Precision–Recall (P–R) curve, P is used to measure the false detection rate of the model and R is used to measure the true detection rate, the formulas for P and R are given in Equations (1) and (2).

Additionally, the metric Intersection over Union (IOU) was also calculated. IoU is a fundamental metric in object detection that measures the degree of overlap between the predicted bounding box (generated by the algorithm) and the ground truth bounding box (annotated using labeling software), with a range of values from 0 to 1. A higher IoU value signifies better prediction accuracy, representing a greater overlap between the predicted bounding box and the ground truth bounding box. The formula for calculating IoU is as follows:

$$IoU = \frac{A \cap B}{A \cup B} \tag{5}$$

where A denotes the area of the detection box, while B refers to the area of the ground truth box.

IoU is widely used to determine whether a predicted bounding box is considered a true positive. As the output quality of the model varies with changes in the Intersection over Union (IoU) threshold, it is standard practice to evaluate model performance across multiple IoU thresholds.

mAP50 and mAP50-95 are core metrics for evaluating object detection model performance, each measuring the model's average detection precision under different IoU threshold conditions. mAP50

indicates the average precision computed at an IoU threshold of 0.5, where an overlap of over 50% between
the detection box and ground truth box qualifies as a correct detection. This metric primarily reflects the
model's basic object detection capability. mAP50-95, on the other hand, is the mean average precision
calculated across ten different IoU thresholds, from 0.5 to 0.95 in 0.05 increments. This metric averages AP
across multiple IoU thresholds, providing a comprehensive assessment of the model's performance. A higher
mAP50-95 value indicates stronger generalization ability across varying degrees of overlap within the same
scene. Collectively, mAP50 and mAP50-95 provide a thorough and objective evaluation of model prediction
accuracy and are extensively utilized in the domain of object detection.

## 311  3. Result

### 312  3.1 Training experimental results

### 313  3.1.1 Analysis of training results on the original dataset

This subsection provides a comparative evaluation of the training outcomes of the different YOLOv8
models on the original dataset. Each model was trained under the experimental settings outlined in Section
3.1. Fig. 5 illustrates the trend of mAP for each model during the training period, while Fig. 6 presents the
prediction performance of each model across individual classes.
As illustrated in Fig. 5a, following 100 epochs of training, the four models (YOLOv8n, YOLOv8s,
YOLOv8m, YOLOv8l) each reach convergence, achieving mAP50 values on the validation dataset of 0.618,
0.662, 0.611, and 0.639, respectively. YOLOv8s exhibits superior performance in this object detection task,
maintaining a higher mAP50 across the training duration compared to the other configurations. YOLOv8l
shows pronounced oscillations during the early training stages, requiring a longer stabilization period.
Furthermore, it is observable that mAP50 for all models increases rapidly in the early epochs (0-20), after
which the growth rate decelerates and plateaus. Notably, an increase in network width (param) and depth
(layer) for YOLOv8 does not yield a substantial improvement in mAP50.

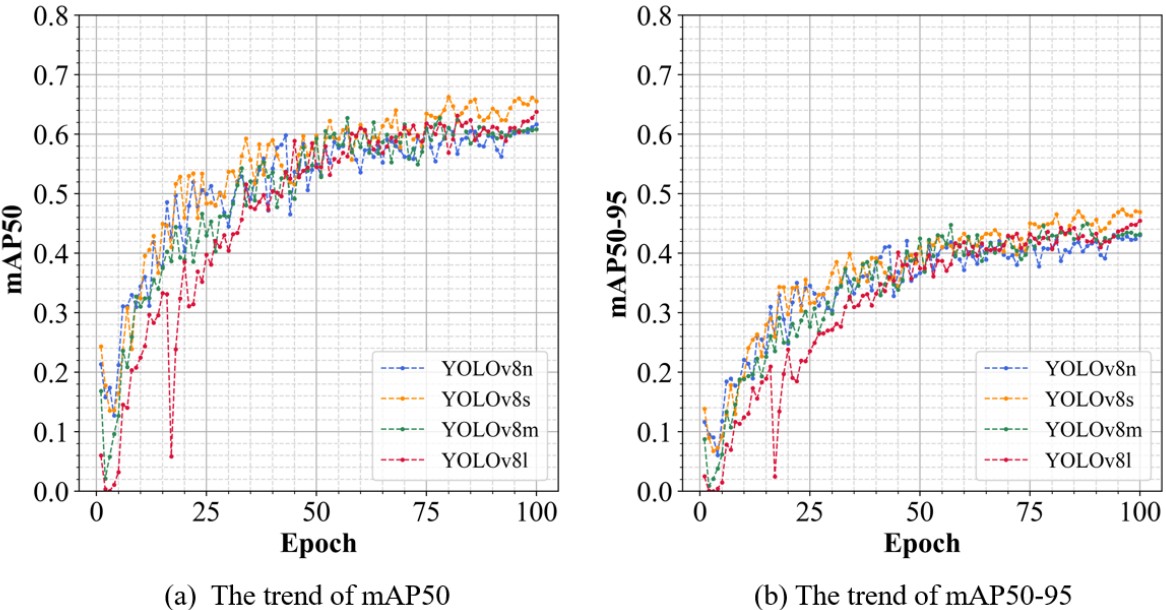

(a) The trend of mAP50                                  (b) The trend of mAP50-95

**Fig. 5.** The mAP of the validation dataset throughout the training process (a) The trend of mAP50; (b) The trend of mAP50-95.

As shown in Fig. 6, the detection performance of the four YOLOv8 models varies across different flood risk levels. The YOLOv8s model demonstrates superior average precision across all flood risk categories, with an mAP50 of 0.662. For low flood risk category identification (level 1), all models achieve satisfactory detection results. However, for higher flood risk categories, the detection performance of YOLOv8n, YOLOv8m, and YOLOv8l is below expectations, with only YOLOv8s effectively capturing the submersion characteristics of buses at high water levels, thereby achieving relatively accurate identification. Although the YOLOv8l model has higher complexity and a larger parameter scale, its AP values for the level 3 and level 4 categories are 0.591 and 0.530, respectively, significantly lower than those of the YOLOv8s model, which are 0.632 and 0.602. This suggests that, for detecting high flood risk features, increased model complexity does not inherently result in improved detection accuracy.

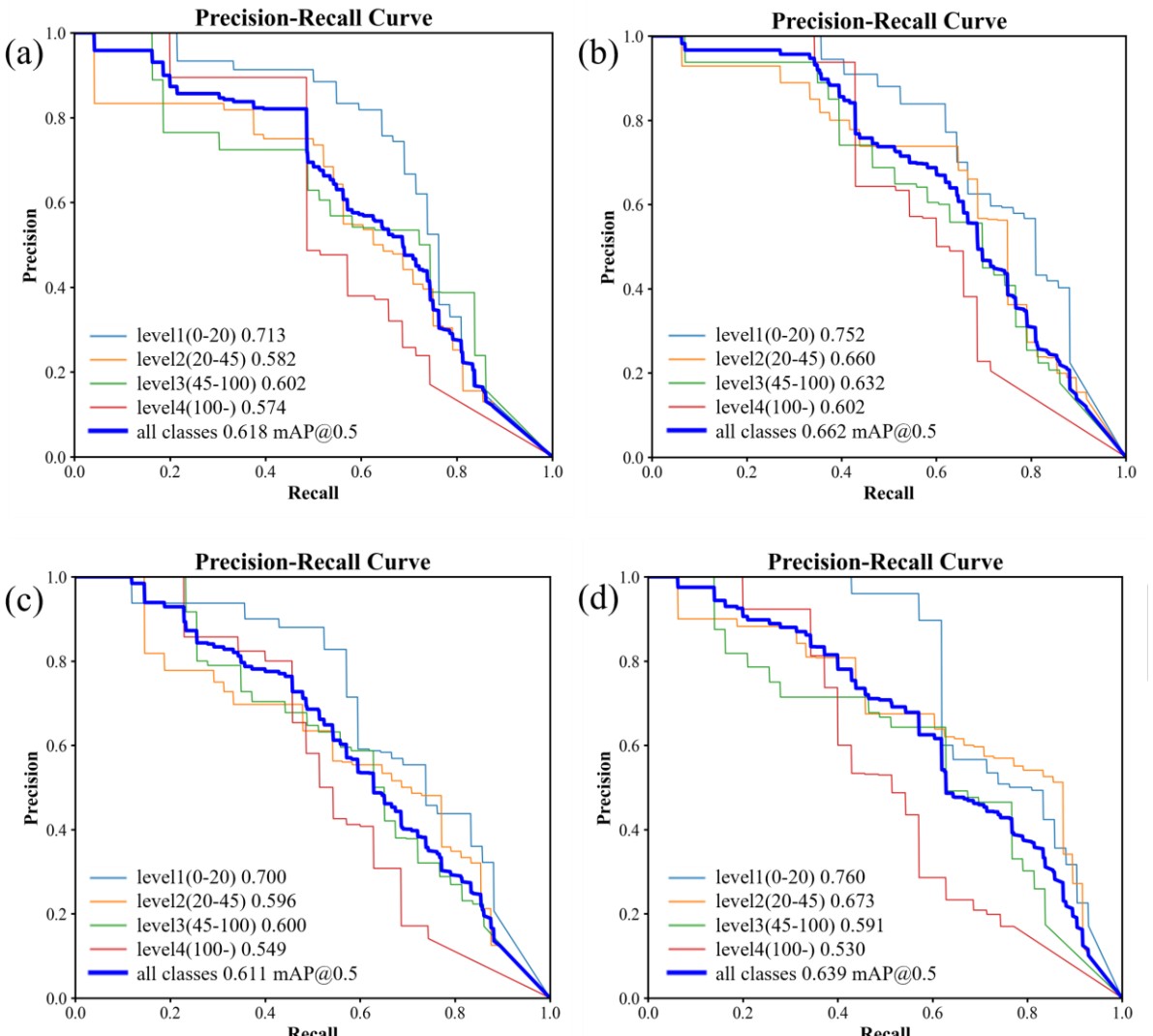

**Fig. 6.** Precision-recall curve for each class of validation process: (a) YOLOv8n Validation Results; (b)
YOLOv8s Validation Results; (c) YOLOv8m Validation Results; (d) YOLOv8l Validation Results.

**3.1.2 Analysis of training results on the augmented dataset**

This subsection compares the training performance of different YOLOv8 models on the augmented
dataset. Fig. 7 illustrates the mAP trend, while Fig. 8 presents the prediction performance of each model.
As depicted in Fig. 7a, after 100 training epochs, the four models (YOLOv8n, YOLOv8s, YOLOv8m,
YOLOv8l) all converge, reaching mAP50 values on the validation dataset of 0.722, 0.734, 0.716, and 0.703,
respectively. Among them, YOLOv8s consistently outperforms the other models. The early-stage
fluctuations observed during YOLOv8l training disappear, resulting in smoother curves and a more stable
optimization process. However, compared to other models, the performance improvement of YOLOv8l
during the early stages of training remains relatively slower. This may be attributed to the model's higher
complexity and larger parameter scale.

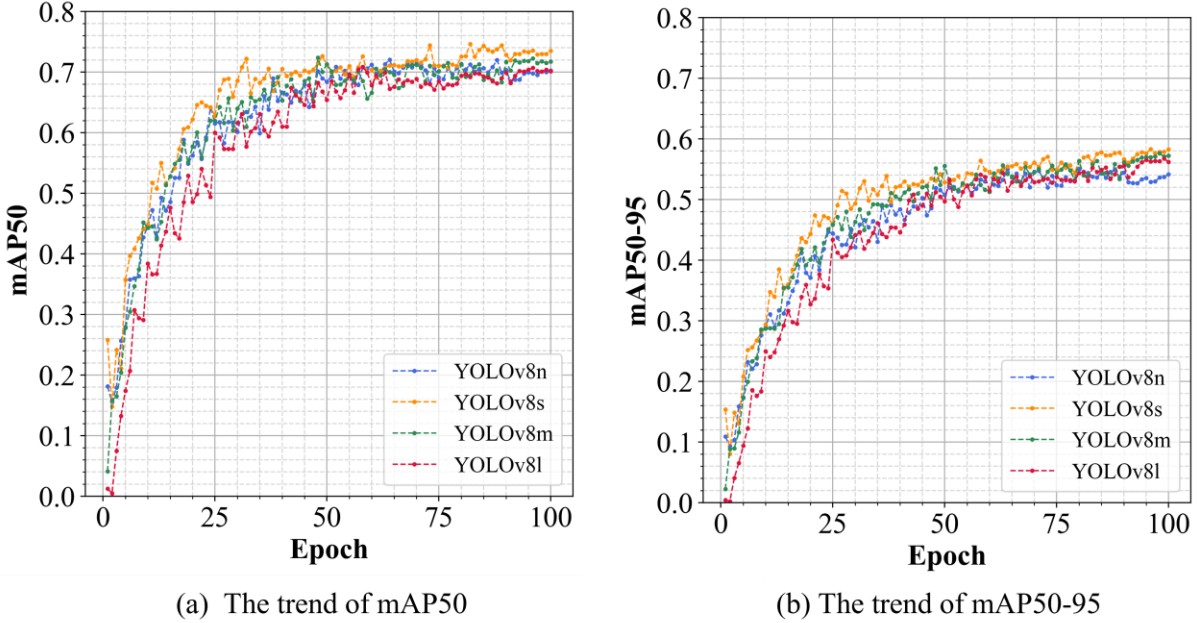

(a) The trend of mAP50         (b) The trend of mAP50-95

**Fig. 7.** The mAP of the validation dataset throughout the training process after data augmentation: (a) The trend of mAP50; (b) The trend of mAP50-95.

As illustrated in Fig. 8, following data augmentation, the four YOLOv8 models exhibit similar performance in detecting each flood risk level. The highest overall effectiveness is found in YOLOv8s, achieving an mAP50 of 0.734, with consistently high precision and recall across all flood risk categories. All models effectively capture the submergence characteristics of buses at various water levels, with AP values for each category exceeding 0.67, indicating robust performance on the validation dataset.

Notably, the detection results for higher-risk categories (Level 3 and Level 4) show improved AP values in all models trained with augmented images, as evidenced by the Precision-Recall curves shifting closer to the upper-right corner. For example, in the YOLOv8n model, the AP value for Level 3 increased from 0.602 to 0.712, and for Level 4, from 0.574 to 0.746. Similarly, in the YOLOv8l model, the AP value for Level 3 rose from 0.600 to 0.685, and for Level 4, from 0.549 to 0.679. This improvement is primarily attributed to enhanced data diversity, effectively mitigating the interference caused by variations in viewpoint and object scale. This not only improved the model's ability to detect higher-level targets but also enhanced the overall detection performance.

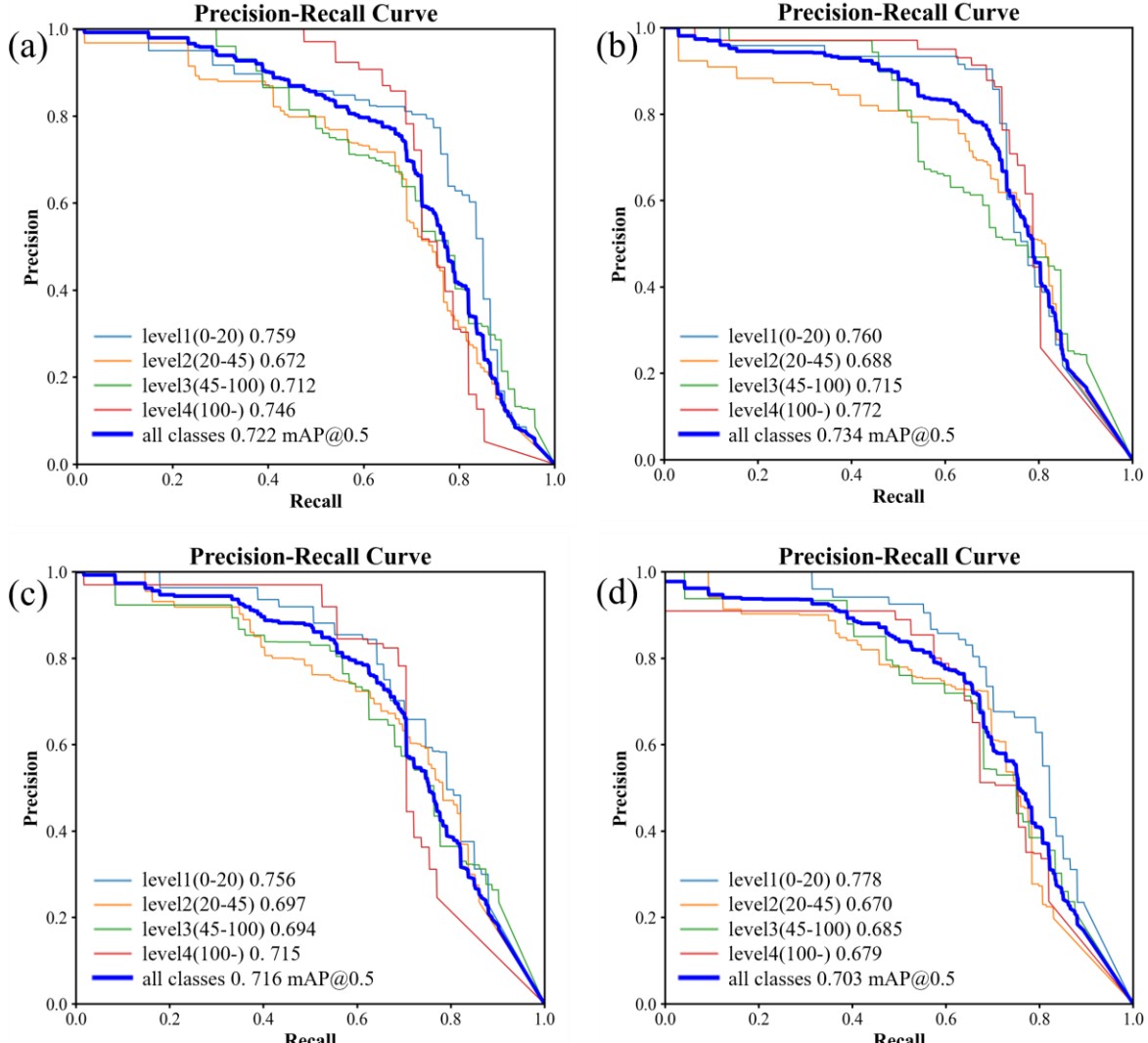

**Fig. 8.** Precision-recall curve for each class of validation process after data augmentation: (a) YOLOv8n
Validation Results; (b) YOLOv8s Validation Results; (c) YOLOv8m Validation Results; (d) YOLOv8l
Validation Results.
This study provides a comparative analysis of the detection performance of different YOLOv8 model
configurations before and after data augmentation. Table 4 presents the mAP50, mAP50-95, and training
times for the optimal YOLOv8 models. Table 5 further details the mAP50-95 performance of the four models
across different flood risk levels.
**Table 4**
mAP and training time for four YOLOv8 configurations: pre- and post-data augmentation.

| YOLO v8 | mAP50 (%) | | | mAP50-95 (%) | | | training time (h) | |
|---|---|---|---|---|---|---|---|---|
| | original dataset | augmented dataset | increase | original dataset | augmented dataset | increase | original dataset | augmented dataset |
| n | 61.7 | 72.2 | 17.0 | 43.0 | 55.2 | 28.4 | 0.102 | 0.325 |
| s | 66.0 | 73.4 | 11.2 | 47.5 | 58.2 | 22.5 | 0.152 | 0.483 |
| m | 61.2 | 71.6 | 17.0 | 45.0 | 57.4 | 27.6 | 0.293 | 0.930 |
| l | 63.8 | 70.3 | 12.2 | 45.4 | 56.6 | 24.7 | 0.438 | 1.390 |

As indicated in Table 4, improvements in detection performance were observed across all model
configurations when trained on the augmented dataset. For mAP50, all models demonstrated improvements

exceeding 10%, while mAP50-95 gains surpassed 20% for each configuration. YOLOv8m and YOLOv8n

exhibited the highest mAP50 increases, whereas YOLOv8s showed the smallest gain at 11.2%. In terms of

mAP50-95, YOLOv8n achieved the greatest improvement, rising from 43.0% to 55.2% (28.4% increase),

while YOLOv8s displayed the smallest increase at 22.5%.

In terms of overall performance, YOLOv8s achieved the highest mAP50 and mAP50-95 values across

all configurations, reaching 73.4% and 58.2%, respectively, with a moderate training time. By contrast,

YOLOv8l achieved similar mAP50 and mAP50-95 values but required nearly triple the training time.

Similarly, YOLOv8m showed performance close to that of YOLOv8s, though with a nearly doubled training

time.

**Table 5**

improvement in mAP50-95 of YOLOv8 Models across risk levels after data augmentation

| Flood Levels | Model | Original dataset | Augmented dataset | Increase(%) |
| --- | --- | --- | --- | --- |
| Level1 | n | 0.504 | 0.597 | 18.5 |
| | s | 0.520 | 0.640 | 23.1 |
| | m | 0.491 | 0.609 | 24.0 |
| | l | 0.540 | 0.636 | 17.8 |
| Level2 | n | 0.377 | 0.523 | 38.7 |
| | s | 0.492 | 0.555 | 12.8 |
| | m | 0.476 | 0.586 | 23.1 |
| | l | 0.505 | 0.548 | 8.5 |
| Level3 | n | 0.410 | 0.500 | 22.0 |
| | s | 0.465 | 0.504 | 8.4 |
| | m | 0.434 | 0.505 | 16.4 |
| | l | 0.429 | 0.524 | 22.1 |
| Level4 | n | 0.411 | 0.592 | 44.0 |
| | s | 0.423 | 0.625 | 47.8 |
| | m | 0.410 | 0.590 | 43.9 |
| | l | 0.339 | 0.548 | 61.7 |

As shown in Table 5, all four YOLOv8 variants achieved measurable improvements in mAP50-95

across all flood risk levels following data augmentation, indicating enhanced detection accuracy. The most

substantial gains occurred under the highest risk level (Level4), where YOLOv8l exhibited the largest

relative increase—rising from 0.339 to 0.548 (a 61.7% gain). YOLOv8s and YOLOv8m also showed notable

improvements at this level, with increases of 47.8% and 43.9%, respectively. In contrast, improvements

under Levels 1 to 3 were relatively modest. Moreover, YOLOv8n and YOLOv8m demonstrated consistently

stable enhancements across all risk levels. By comparison, YOLOv8s and YOLOv8l displayed minimal

gains at specific levels, possibly due to feature saturation—where the models had already learned most

discriminative patterns, leaving limited room for further improvement through augmentation.. Improvements

in the mAP50-95 metric indicate that the model maintains high detection accuracy even under stricter

evaluation thresholds, reflecting stronger robustness and fine-grained recognition capabilities.
**3.2 Experimental results in complex scenes**
**3.2.1 Validation results of models on the original dataset**
This section presents a comparative evaluation of the validation outcomes for the four optimized

YOLOv8 models trained on the original dataset. The numbers in the top-right corner of the detection boxes
(ranging from 0 to 1) represent the confidence level of the model for the detection results. As shown in Fig.
9, all four models can detect the flood risk level of the bus. However, only YOLOv8l correctly identifies it
as level 1, albeit with low confidence, while the other models incorrectly classify it as level 2. At low flood
risk levels, none of the models effectively capture the critical distinguishing features, highlighting limitations
in their generalization capabilities.

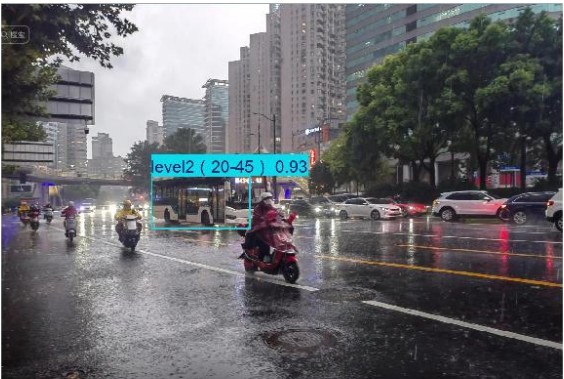 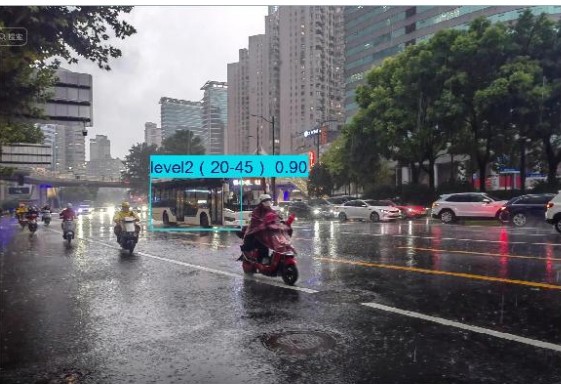

(a) Detection outcomes obtained using YOLOv8n    (b) Detection outcomes obtained using YOLOv8s

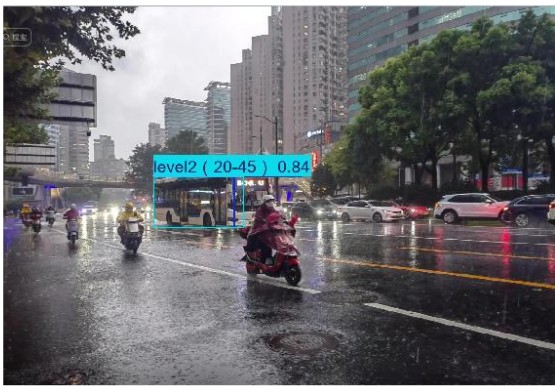 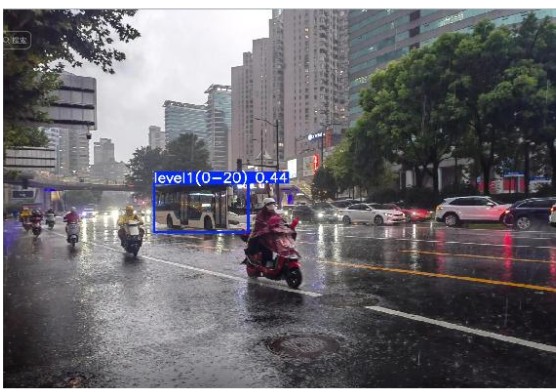

(c) Detection outcomes obtained using YOLOv8m   (d) Detection outcomes obtained using YOLOv8l
**Fig. 9.** Comparison of YOLOv8 detection results in low flood risk scene with multiple vehicles present (pre-
augmentation): (a) Detection outcomes obtained using YOLOv8n; (b) Detection outcomes obtained using
YOLOv8s; (c) Detection outcomes obtained using YOLOv8m; (d) Detection outcomes obtained using
YOLOv8l.

Moreover, Fig. 10 presents the validation results at a high flood risk level. All four optimal YOLOv8

models successfully detect the blurred bus in the image and identify its submerged state with high confidence.
For the bus located at the upper-right edge of the image, none of the models detected it due to low lighting,
though they were able to make partial level assessments. The models have learned to extract key features
associated with incomplete bus bodies from the dataset. Furthermore, YOLOv8l incorrectly classified the
bus station and surrounding environment as level 4 submersion.

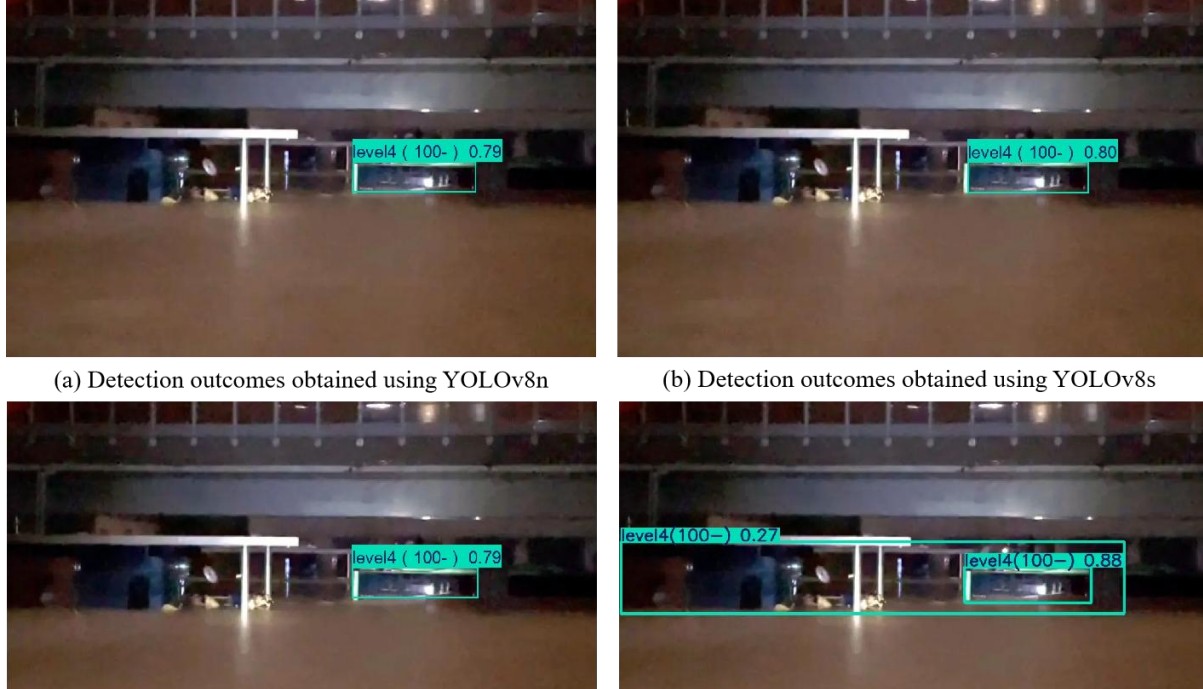

(a) Detection outcomes obtained using YOLOv8n   (b) Detection outcomes obtained using YOLOv8s

(c) Detection outcomes obtained using YOLOv8m   (d) Detection outcomes obtained using YOLOv8l

**Fig. 10.** Comparison of YOLOv8 detection results in high flood risk scene with blurring and corruption (pre-
augmentation): (a) Detection outcomes obtained using YOLOv8n; (b) Detection outcomes obtained using
YOLOv8s; (c) Detection outcomes obtained using YOLOv8m; (d) Detection outcomes obtained using
YOLOv8l.
**3.2.2 Validation results of models on the augmented dataset**
This section provides a comparative evaluation of the validation outcomes for the four optimized
YOLOv8 models after data augmentation. As shown in Fig. 11, all four models accurately detect the flood
risk level of the bus with high confidence. In this detection, the best performance is found inYOLOv8n,
exhibiting higher confidence levels than the other models. Besides, both YOLOv8s and YOLOv8m
unexpectedly generated a level 2 prediction box during bus detection.

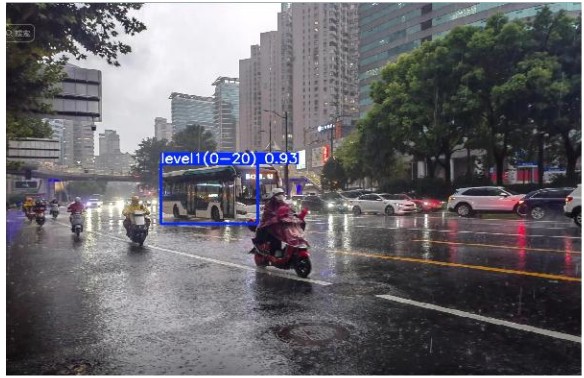 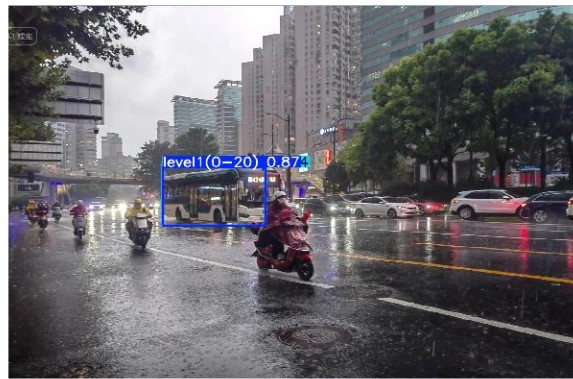

(a) Detection outcomes obtained using YOLOv8n                (b) Detection outcomes obtained using YOLOv8s

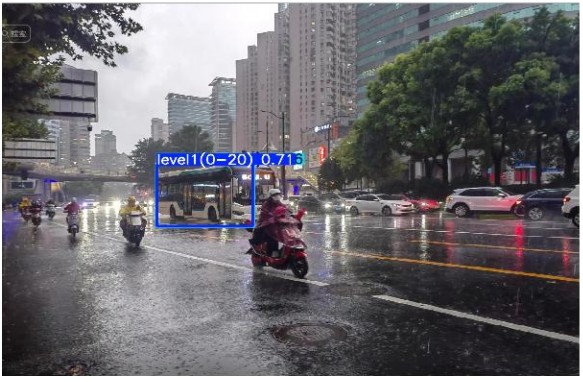

(c) Detection outcomes obtained using YOLOv8m                (d) Detection outcomes obtained using YOLOv8l

**Fig. 11.** Comparison of YOLOv8 detection results in low flood risk scene with multiple vehicles present (post-augmentation): (a) Detection outcomes obtained using YOLOv8n; (b) Detection outcomes obtained using YOLOv8s; (c) Detection outcomes obtained using YOLOv8m; (d) Detection outcomes obtained using YOLOv8l.

Fig. 12 presents the validation results at a high flood risk level. All four optimal YOLOv8 models successfully detect the blurred bus in the image, accurately identifying its submerged state with high confidence. The highest confidence is found in YOLOv8s, without any false detections. However, none of the models detected the bus body located at the upper-right edge of the image, even after data augmentation. Besides, YOLOv8m and YOLOv8l incorrectly classified the bus station and background environment as level 4 submersion.

In summary, improvements in recognition accuracy and confidence were observed across the four optimal models when trained with augmented data, supporting more reliable detection of bus submersion states. In addition, detection performance under low-light conditions remains an area that warrants further investigation.

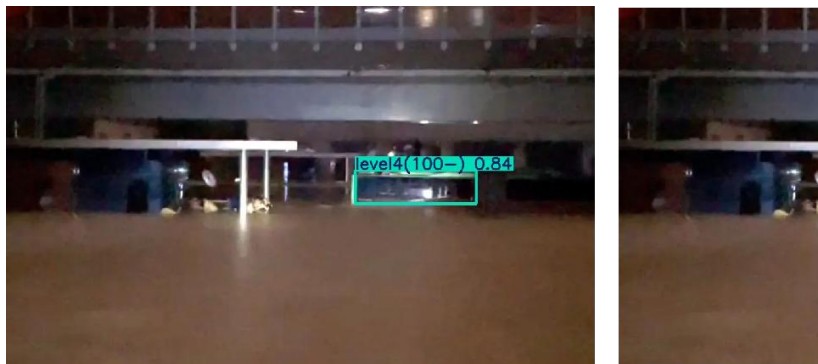

(a) Detection outcomes obtained using YOLOv8n          (b) Detection outcomes obtained using YOLOv8s

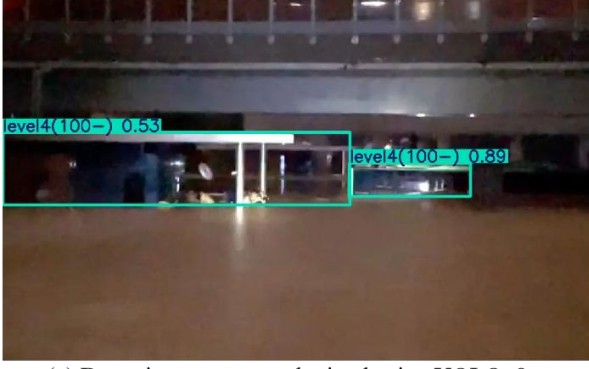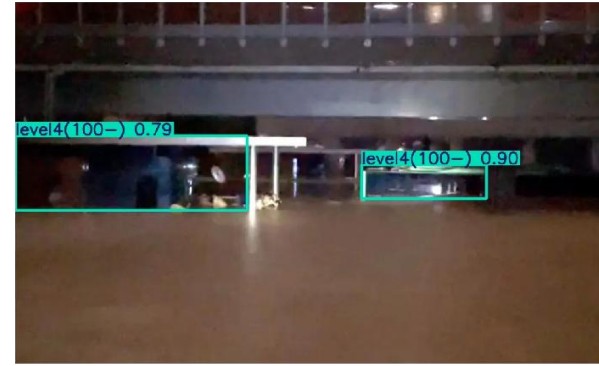

(c) Detection outcomes obtained using YOLOv8m          (d) Detection outcomes obtained using YOLOv8l

**Fig. 12.** Comparison of YOLOv8 detection results in high flood risk scene with blurring and corruption (pre-augmentation): (a) Detection outcomes obtained using YOLOv8n; (b) Detection outcomes obtained using YOLOv8s; (c) Detection outcomes obtained using YOLOv8m; (d) Detection outcomes obtained using YOLOv8l.

### 3.3 Experimental results compared with YOLOv5

### 3.3.1 Analysis of training experiment results

Fig. 13 displays the precision-recall curves from the training results. A comparison reveals that YOLOv8s outperforms YOLOv5s on both the original and augmented datasets, indicating that YOLOv8s achieves higher detection accuracy in image recognition tasks. On the original dataset, YOLOv5s attains an mAP50 of 0.555, while YOLOv8s reaches 0.662, reflecting a performance gap of 19.3%. After data augmentation, although the performance gap between YOLOv5s and YOLOv8s narrows, YOLOv8s continues to lead. Additionally, results from the two training experiments with YOLOv5 indicate that data augmentation has a substantial impact on enhancing the training effectiveness of YOLO models

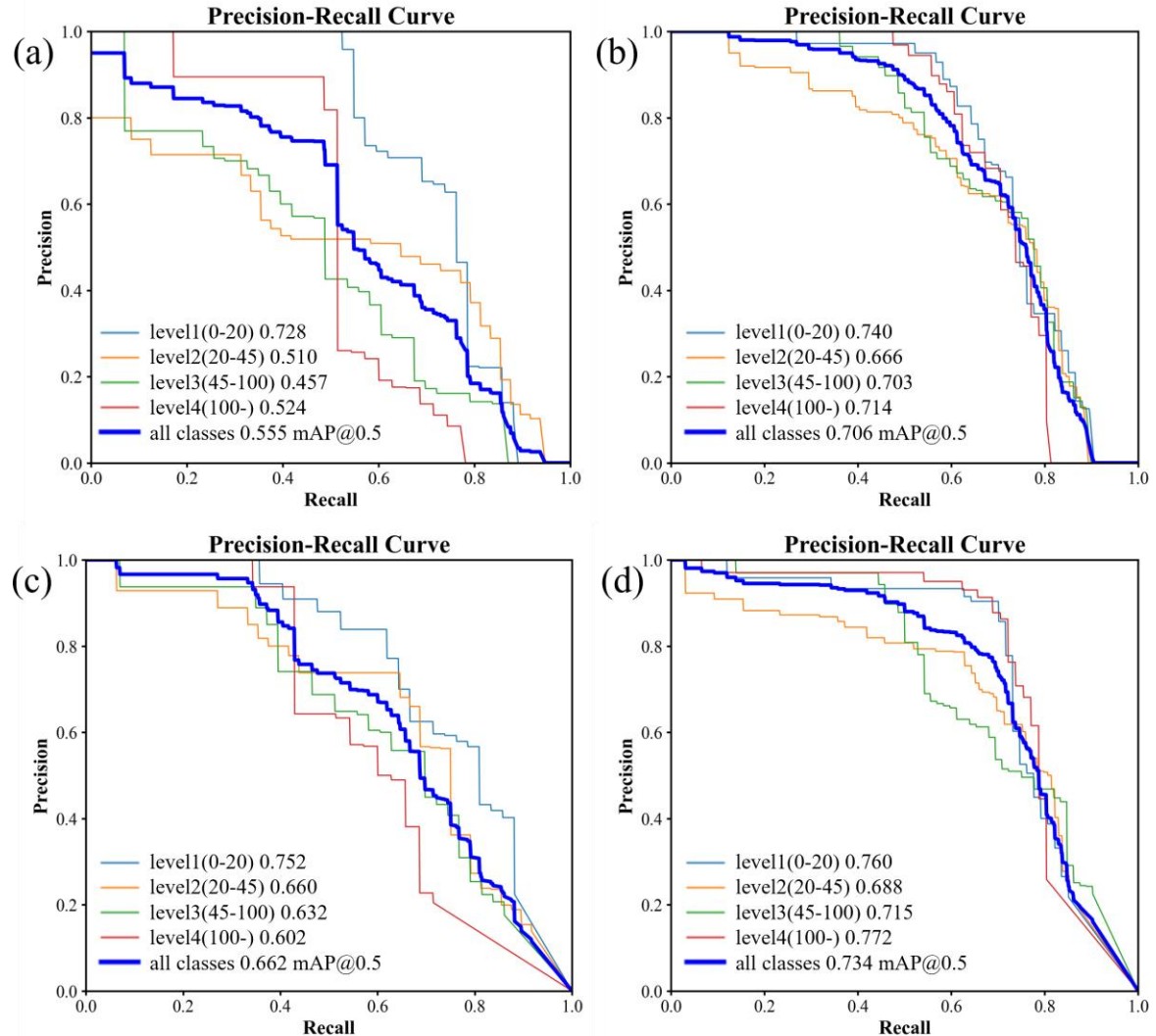

**Fig. 13.** Precision-recall curves from the training results: (a) YOLOv5s validation results on original dataset;
(b) YOLOv5s validation results on augmented dataset; (c) YOLOv8s validation results on original dataset;
(d) YOLOv8s validation results on augmented dataset.
**3.3.2 Analysis of scene prediction experiment results**
This section evaluates the stability and effectiveness of YOLOv5s and YOLOv8s under challenging
conditions by testing their performance in two complex urban scenes. Fig. 14 and Fig. 15 illustrate the
detection results of YOLOv5s and YOLOv8s in low and high flood risk scenes, respectively. In the low-risk
scene, YOLOv5s accurately identifies the bus and correctly predicts its submersion status, similar to
YOLOv8s; however, the confidence level of YOLOv5s, even after data augmentation, shows limited
improvement. YOLOv8s, on the other hand, demonstrates higher detection confidence and accuracy
following data augmentation. In the high-risk scene, characterized by blurred or partially degraded images,
YOLOv5s performs notably well. YOLOv5s detects targets that YOLOv8s fails to recognize, likely due to
its capacity for handling noisy data. This difference in performance under extreme conditions suggests that
the network structure of YOLOv5s may enhance detection in low-quality images, offering insights for
potential optimizations in YOLOv8s.

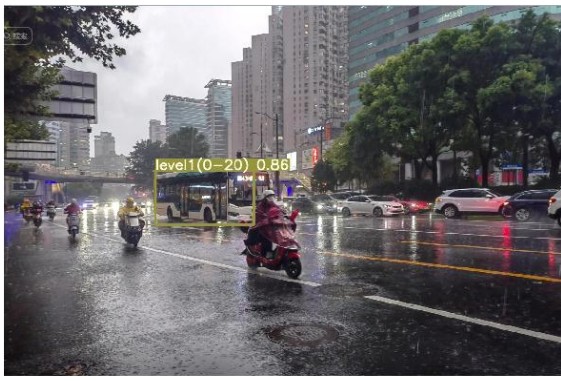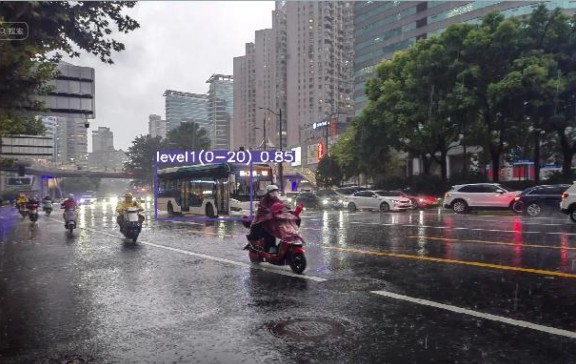

(a) YOLOv5s detection results on original dataset    (b) YOLOv5s detection results on augmented dataset

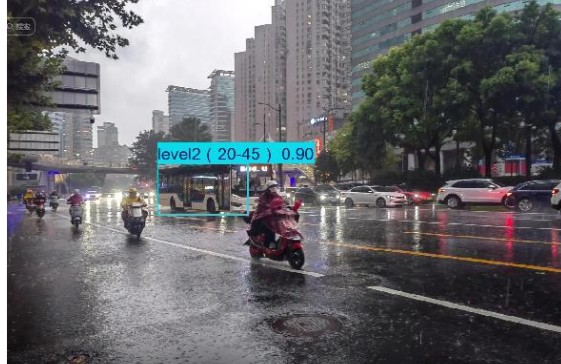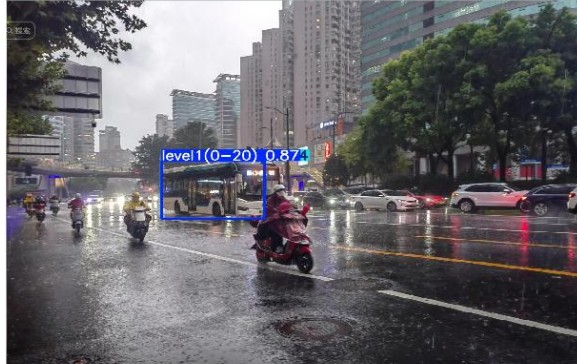

(c) YOLOv8s detection results on original dataset    (d) YOLOv8s detection results on augmented dataset

**Fig. 14.** Comparison of YOLO detection results in low flood risk scene with multiple vehicles present: (a)
YOLOv5s detection results on original dataset; (b) YOLOv5s detection results on augmented dataset; (c)
YOLOv8s detection results on original dataset; (d) YOLOv8s detection results on augmented dataset.

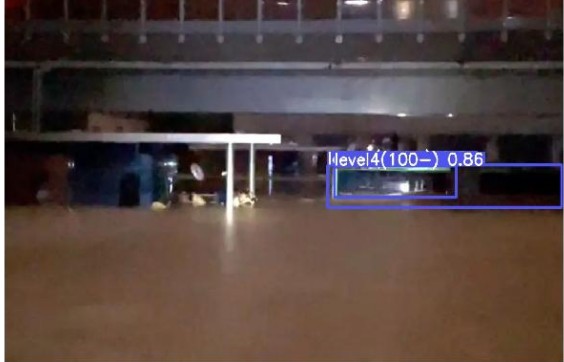 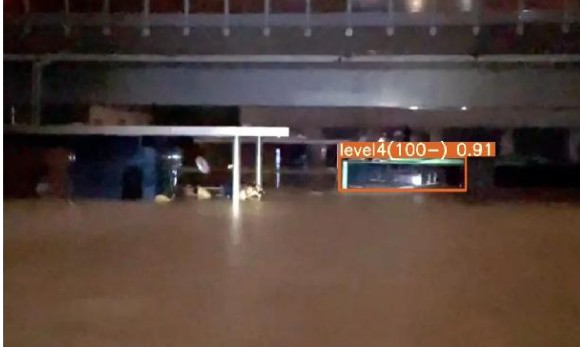

(a) YOLOv5s detection results on original dataset    (b) YOLOv5s detection results on augmented dataset

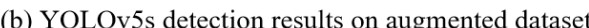

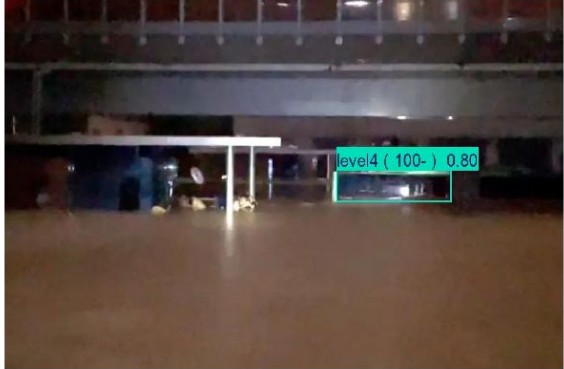 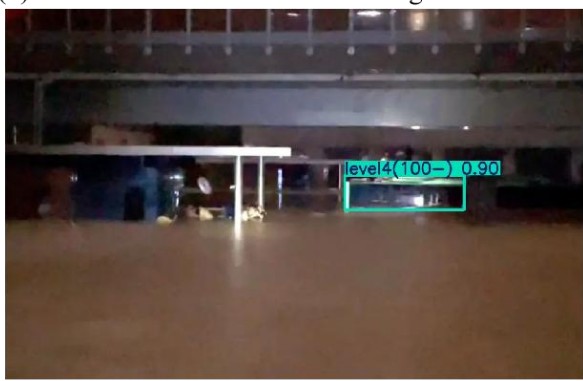

(c) YOLOv8s detection results on original dataset    (d) YOLOv8s detection results on augmented dataset

**Fig. 15.** Comparison of YOLO detection results in high flood risk scene with blurring and corruption: (a) YOLOv5s detection results on original dataset; (b) YOLOv5s detection results on augmented dataset; (c) YOLOv8s detection results on original dataset; (d) YOLOv8s detection results on augmented dataset.

## 4. Discussion

### 4.1 Impact of dataset size and diversity

The size and diversity of the training dataset play a critical role in determining the performance of YOLOv8 models. Models trained with the augmented dataset achieved substantial improvements, with mAP50 and mAP50-95 on the validation set increasing by over 10% and 20% (Table 4), across all YOLOv8 configurations. The increased number and diversity of training samples not only enable a more comprehensive understanding of the key features of detection targets but also allow for a broader representation of annotated instances, enhancing the model's ability to learn. Such exposure to varied input scenarios equips YOLOv8 with enhanced generalization, leading to better results in the validation stage.

Moreover, this benefit extends beyond training to inference capabilities. Under challenging conditions such as low illumination, motion blur, and partial occlusion, models trained on the augmented dataset exhibited greater robustness, reflected in higher detection confidence. Dataset diversity is therefore key to improving training performance and significantly enhancing inference in complex real-world environments.

The impact of dataset size and diversity varies across different YOLOv8 configurations. YOLOv8n,

with the smallest number of parameters, is particularly responsive to changes in training sample volume,
showing the most notable improvement in mAP when trained on the augmented dataset. In contrast,
YOLOv8s, which already performed well on the original dataset, shows the smallest relative gain.
Nevertheless, it consistently delivers the best results across multiple experiments.
**4.2 Recommended configurations for YOLOv8**
Based on the comprehensive performance of different YOLOv8 configurations during training and in
complex scenes, this study recommends prioritizing the YOLOv8s model for urban flood detection using
bus imagery. Although the YOLOv8l model theoretically offers higher network complexity and parameter
count, which should enable more granular feature extraction, its performance improvements in this
experiment were not significant. In contrast, the YOLOv8s model demonstrated superior results across
multiple performance metrics and achieved an effective balance between model accuracy, training time, and
computational resource requirements. Therefore, in environments with limited resources and high
computational costs, YOLOv8s has proven to be the most advantageous choice.
When computational resources and dataset availability are ample, YOLOv8m or YOLOv8l should be
considered as priority options. As dataset size expands and more computational resources become available,
the performance potential of YOLOv8m and YOLOv8l models may be more fully realized. With larger
datasets, the deeper network structures and advanced feature extraction capabilities of YOLOv8m and
YOLOv8l can better capture critical features and details of the targets, resulting in higher accuracy and
stability in object detection.
**4.3 Advantages and disadvantages**
Crowdsourced images from social media have emerged as a high-value data source for acquiring flood
information (Huang et al., 2020). The YOLO object detection model requires annotated images as input for
training, and such annotation can be performed by non-experts, making the method highly scalable. Buses,
as structurally standardized and dimensionally stable vehicles, provide reliable and scalable reference objects
for flood status identification. Moreover, buses typically operate in traffic-intensive or critical urban areas,
where water-related safety concerns directly impact the stability of the public transportation system. The bus
submergence detection technique adopted in this study significantly enhances the precision of urban flood
risk assessment, offering a more scientific basis for future flood monitoring and emergency response in cities.

Compared with existing studies, the method proposed in this paper demonstrates distinct advantages in

terms of reference object selection and object detection algorithms. The present study can be compared with
these approaches as follows: (Jiang et al., 2020), (Bhola et al., 2018) and (Alizadeh Kharazi and Behzadan,
2021) respectively utilized traffic cones and guardrails, bridge structures, and sign poles as reference
indicators. Although these objects offer advantages such as fixed structure and known physical dimensions,
their applicability is often confined to specific locations with fixed viewing angles and clearly visible
reference targets. In contrast, this study adopts buses as reference objects. Given their extensive operating
routes and easily recognizable structural features, the proposed method enables flood perception across
broader urban road networks, thereby exhibiting enhanced spatiotemporal adaptability.

(Huang et al., 2020) and (Park et al., 2021) applied the Mask R-CNN model to detect vehicle tires or

body structures to support flood level estimation. While their methods achieve high prediction accuracy,
Mask R-CNN is a two-stage detection framework with a complex architecture and relatively slow inference
speed. Additionally, it is highly sensitive to image quality and susceptible to errors under occlusion, water
splashes, or reflection interference—conditions common in real-world flood scenarios—which limits its
real-time applicability and environmental robustness (Huang et al., 2020). By contrast, the YOLOv8 model
adopted in this study is a single-stage, end-to-end detection framework. It demonstrates superior robustness
and detection speed under complex scene conditions and can stably identify reference targets, reducing false
detection rates. Its lightweight architecture also facilitates deployment on edge devices (Liu et al., 2025),
making it suitable for practical engineering applications.

(Wan et al., 2024) employed YOLOv8 to classify flood submergence levels of cars and achieved the

highest mAP50 of 0.707, demonstrating satisfactory detection accuracy. In their complex scenario testing,
sedans could be correctly identified, whereas pickup trucks failed to be recognized. This observation indicates
that the diversity of vehicle features may affect the model's generalization capability. In comparison, the
structural consistency and height stability of buses help reduce the impact of target variability on model
performance. Moreover, the best-performing model in this study achieved a mAP50 of 0.734, demonstrating
superior detection accuracy.

This study demonstrates that floodwater levels in urban areas can be effectively identified by detecting

the submergence status of buses using the YOLOv8 model. The proposed method has been validated in
terms of feasibility and practical potential. However, further research is warranted to enhance the model's
adaptability and applicability. On the one hand, the current dataset remains limited in scale, with insufficient
images under nighttime or low-light conditions, which constrains the model's performance under such
scenarios. Future work may incorporate illumination-guided transformers or light-aware attention
mechanisms (Cai et al., 2023) to improve model robustness under challenging lighting environments.
Simultaneously, expanding the data acquisition scope and incorporating more diverse scenes will be critical
for improving detection accuracy. On the other hand, the current detection framework does not support
quantitative estimation of floodwater depth. Future research could explore regression-based modules that
leverage the actual height of buses and the geometric relationships within images to develop numerically
interpretable flood level estimators, thereby providing more actionable support for urban emergency
response and flood risk management.
**5. Conclusions**
This study proposes an urban flood detection method based on the YOLOv8 deep learning model, which
accurately assesses flood risk levels by identifying the submersion state of buses. A dataset of 1,008 images
depicting submerged buses was collected from online platforms and expanded to 2,184 images using data
augmentation strategies. Subsequently, the submersion states of buses were annotated into four levels for
training the object detection model. Finally, the performance of various YOLOv8 models was compared
through data augmentation and complex scene validation experiments, resulting in the following
experimental insights:
1)   The highest detection accuracy for flood risk levels is achieved by YOLOv8s. Although YOLOv8m

and YOLOv8l demonstrated comparable overall performance, they required significantly greater

computational resources and training time.

2)   Compared with commonly used reference objects in existing studies, buses offer advantages such as

structural standardization, consistent height, and wide spatial coverage, making them more suitable for

flood identification at the urban scale.

3)   This study offers configuration recommendations for YOLOv8 models tailored to urban flood

detection based on the submersion state of buses.

This study supplements and extends existing research on flood detection by validating the feasibility
and effectiveness of using buses as reference objects for the identification of urban flood risk levels. This
study explores the application potential of bus submersion state detection within the YOLOv8 framework.
This approach broadens the technical pathways for urban flood monitoring and provides crucial support for
flood emergency management within urban transportation systems, demonstrating practical value in the field
of disaster prevention and mitigation.

This study primarily relies on social media image recognition to determine flood risk levels. However,

the model still lacks robustness in certain complex scenarios, such as under extreme lighting conditions.
Additionally, the model currently lacks the capability to quantitatively measure flood depth, which is critical
for precise flood risk assessment. Future research will concentrate on devising methods for quantitative flood
depth estimation using bus submersion states and creating noise-resistant model architectures to improve the
model's applicability and precision.
**Data availability:**

Data will be made available on request.

**Declaration of competing interest**

The authors declare that they have no known competing financial interests or personal

relationships that could have appeared to influence the work reported in this paper.
**Computer Code Availability:**

The code in this study can be obtained from the Git repository:

https://github.com/ydkyly/dataset_and_result.git.

**Authorship Statement**
**YQ:** Writing – original draft, Validation, Software, Methodology, Investigation. **XZ:** Writing – review &
editing, Validation. **JW:** Writing – review & editing, Project administration, Validation. **TY:** Writing –
review & editing, Supervision. **LZ:** Formal analysis, Validation. **YZ:** Data curation, Validation. **LS:** Data
curation, Validation. **XJ:** Validation.

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
