# Peer review of "Automated urban flood level detection based on flooded bus dataset using YOLOv8"

_EGUsphere, 2024_

## Author Response (AR1)

**Title:** *Automated urban flood level detection based on flooded bus dataset using YOLOv8*
**Manuscript Number:** *egusphere-2024-4053*
**Authors:** *Yanbin Qiu, Xudong Zhou, Jiaquan Wan, Tao Yang, Lvfei Zhang, Yuanzhuo Zhong, Leqi Shen, and Xinwu Ji*

Dear Referee,

We are truly grateful for your meticulous review and valuable suggestions on our manuscript. Based on your comments, we have revised the manuscript point-by-point and provided detailed explanations for the relevant issues. We have strived to enhance the rigor and clarity of the paper through these improvements and sincerely hope to gain your approval. The specific responses are as follows:

**Referee #1**

**1) Regarding the validation set:**

**Comment:**

*"Can the authors please clarify whether the 10% validation set is synonymous with a 'test set'? Specifically, was this set completely withheld during training—including during data augmentation—such that no original or augmented images in this set were seen by the model?"*

**Response:**

Thank you for your valuable comment. We would like to further clarify that the 10% validation set mentioned in our study can indeed be considered equivalent to a "test set" as you interpreted. Throughout the entire training process, the validation set was strictly separated from the training set, and the model never accessed any original or augmented images from the validation set.

During data processing, data augmentation techniques were applied exclusively to the training set, while the validation set remained as untouched original images. The validation set was used solely for performance evaluation after each training epoch, primarily to monitor the training process, prevent overfitting, and ensure a fair assessment of the model's generalization ability.

**Changes in manuscript:**

In the revised manuscript, we have supplemented relevant explanations on page 10, lines 225–230.

**2) Regarding hyperparameters tuning:**

**Comment:**

*"Did the authors adjust any hyperparameters of the YOLOv8 algorithm? If so, could they describe the tuning process?"*

**Response:**

Thank you for raising this point. In this study, we employed the official default hyper parameters settings of YOLOv8 without any additional tuning. Parameters such as learning rate, batch size, and confidence threshold were maintained at their default values to ensure the consistency and reproducibility of the model results.

**Changes in manuscript:**

In the revised manuscript, we have supplemented relevant explanations on page 10, lines 230–231.

**3)Regarding the evaluation of complex scenes:**

**Comment:**

*"The authors discuss two example images as case studies for 'complex' scenes. Can the authors elaborate whether there is a larger dataset of such complex scenes on which the model performance was evaluated? If not, how were these two specific examples selected? How do the authors anticipate the model will generalize to similar complex scenarios?"*

**Response:**

Thank you for your insightful question regarding the evaluation of complex scenes. Currently, there is no publicly available large-scale dataset of bus flood inundation images, and the images retrieved from social media predominantly depict regular scenes rather than extreme complex scenarios. Due to the scarcity and dispersion of such complex scene images, we were unable to construct an independent large-scale evaluation dataset.

The two examples were selected based on their representative difficulty and relevance to the target application. The selection criteria included: (1) ensuring diversity by covering common interference factors such as low-light nighttime conditions, object occlusion, and multiple object overlaps, thereby avoiding bias caused by a single disturbance type; (2) prioritizing scenes presenting compound challenges that were not

sufficiently covered during training, to better assess the model's adaptability to unseen complex environments.

During the training process, we employed multi-level and diverse data augmentation strategies to significantly increase the complexity and diversity of the training data, encouraging the model to learn more robust feature representations. Experimental results demonstrate that the augmented model achieved notable improvements in complex scene detection. Based on the current results, we believe that the model has good generalization potential and can effectively handle unseen complex scenes. In future work, we plan to further validate the model performance on a larger-scale dataset.

**4) Suggestion to introduce YOLOv8 earlier**

**Comment:**

*"To improve accessibility of the manuscript for a broader audience, I would suggest introducing the YOLOv8 algorithm with a short description earlier in the manuscript."*

**Response:**

We agree with this suggestion and have added a concise description of the YOLOv8 algorithm in the Introduction section to help readers unfamiliar with the model.

**Changes in manuscript:**

We add a brief introduction to YOLOv8 in the Introduction section on page 4, lines 108–117

**5) Specific line edits**

Below we address each line-specific issue and corresponding corrections:

| | Comment | Response | Page & Line(s) |
|---|---|---|---|
| **1** | *Line 14 - have been emerged* | Corrected to *"have emerged"*. | Page 1, Line 14 |
| **2** | *L19 - YOLOv8 is referenced without a preceding description.* | Added a short description | Page 1, Lines 16–17 |
| **3** | *L23 - as they remain *in* service* | Grammar corrected for clarity. | Page 1, Line 25 |
| **4** | *L69 - Park et al. 2021 is cited twice for the same statement.* | Removed duplicate citation. | Page 3, Lines 87–88 |
| **5** | *L70 - Suggest describing YOLO as a CNN-based CV model prior to first usage.* | Added description at first mention. | Page 2, Lines 68–73 |

| | Comment | Response | Page & Line(s) |
|---|---|---|---|
| **6** | *L85 - submerged states of buses *is* categorized* | Corrected to *"are categorized"*. | Page 4, Line 105 |
| **7** | *L88 - Missing source citation* | Added appropriate source. | Page 4, Line 116 |
| **8** | *L101 - configurations, *and* explains the experimental design *and* model evaluation metrics* | Revised for consistent phrasing. | Page 5, Line 129 |
| **9** | *L113 - images in exhibit* | Rephrased to *"images used in the experiment vary in"*. | Page 5, Line 141 |
| **10** | *L122 - What does "instances" refer to?* | The instances labeled with Labelimg refer to the buses and their submersion statuses that have been annotated in the images. | Page 5, Line 159 |
| **11** | *L165 - Suggest expanding acronyms at first usage.* | Acronyms fully expanded at first use. | Page 8 Line 201 |
| **12** | *L189 - Based on the sentence, it appears that the 90-10 split was done after data augmentation, while the correct approach will be to perform data augmentation only on the training set, to avoid data leakage. Please clarify in text.* | Clarified the procedure. | Page 10, Lines 225–230 |
| **13** | *L198 - two particularly demanding *scenes** | Improved wording. | Page 10, Line 239 |
| **14** | *L206 -The statement is unclear.* | Rewritten for clarity. | Page 11, Lines 247–249 |
| **15** | *Eq 4 - The parameters - n, AP, P, R - are undefined.* | Defined all terms in the text. | Page 11, Lines 270–272 |
| **16** | *L228 - Suggest introducing IoU prior to first usage.* | Added short definition at first use. | Page 12, Lines 273–277 |
| **17** | *L426 - all four YOLOv8 models may exhibited* | The revised content has been updated | Page 24, Line 460 |

**Referee #2**

**1) On the Practical Significance of Water Depth Estimation:**

**Comment:**

*"The manuscript should clarify the practical implications of the estimated flood depths. For instance, the difference between 20 cm and 45 cm of water may not significantly affect early disaster response decisions. In contrast, a depth corresponding to the average floor height might indicate a risk of people being trapped and requiring rescue. I recommend the authors provide more context on how the predicted water depths contribute to emergency decision-making."*

**Response:**

We are grateful for your constructive suggestion. The practical significance of water depth estimation in emergency response certainly deserves further clarification. In the original manuscript, our primary focus was on estimating flood water depth distributions from social media images to help identify potential flood risk areas in urban settings.

In practical applications, even a difference of just a few centimeters in water depth can have varying impacts. For example, 20–30 cm of flooding may cause vehicle stalls or obstruct pedestrian movement, while water levels approaching or exceeding 40–50 cm are more likely to enter residential or commercial areas, posing a significant threat to personal safety and property. By referencing objects like buses and bus door steps in the images, we attempt to provide a visual risk indicator. We hope that the predicted depth ranges will assist emergency response teams in assessing flood severity, identifying high-risk areas, and planning priority responses. In the revised manuscript, we will carefully consider your suggestions and provide further clarification on this point. Thank you again for your valuable feedback.

**Changes in manuscript:**

In the revised manuscript, we have supplemented relevant explanations on page 2, lines 43–61.

**2) On Using Buses as Reference Objects:**

**Comment:**

*"The authors argue that using buses as reference objects improves accuracy. However, it is unclear whether potential buoyancy or floating of the buses was considered. Did the authors verify that the buses used as reference points remained stationary during the flood? Also, please specify the assumed bus height used in the model calibration or estimation."*

**Response:**

We appreciate your attention to this issue. During the model development process, we paid particular attention to the reliability of the reference objects. All images used for training and testing were manually screened to ensure that the buses in the images were either stationary or moving normally in the floodwaters, and that there was no buoyancy or floating involved, ensuring that buses remained stable reference points for accurate measurements.

We standardized the height of the buses in our model to approximately 3 meters, referencing typical urban bus models seen in current social media images. Based on a systematic evaluation of the images, we classified flood water depth into several levels, with the majority of images showing water levels between 0–50 cm. Only in a few rare cases did the water level exceed 100 cm. Given the rarity of high-water scenes (i.e., >100 cm) in urban environments, we chose to focus on the more common water levels for classification, without setting higher water levels, as we believe this is more aligned with realistic application needs and image distribution characteristics.

**Changes in manuscript:**

In the revised manuscript, we have supplemented relevant explanations on page 5, lines 146–148 and on page 6, lines 150–157.

**3) On the Description of Data Augmentation:**

**Comment:**

*"While data augmentation is widely recognized to improve model performance, this is already a well-established practice. It may not warrant substantial emphasis in the discussion and conclusions unless the authors offer a novel or particularly insightful implementation."*

**Response:**

Thank you for your comment regarding data augmentation. We plan to modify and simplify this section in the revised manuscript.

**Changes in manuscript:**

Reduced emphasis on augmentation in the discussion and conclusions.

**4) On Providing a Brief Introduction to the YOLO Model:**

**Comment:**

*"The Introduction should include a short explanation of YOLO models, especially considering that not all NHESS readers are familiar with machine learning or object detection frameworks."*

**Response:**

We appreciate your suggestion. We understand that some readers of NHESS may not be very familiar with machine learning and object detection frameworks. Therefore, in the revised manuscript, we will add a brief introduction to the YOLO model in the introduction section, as you suggested.

**Changes in manuscript:**

A brief description of YOLO models has been added in the Introduction (page 4, lines 108–117).

**5) On Established Methods for Estimating Flood Water Depth:**

**Comment:**

*"Are there other established methods for estimating flood depth beyond analyzing social media imagery? If so, a short overview in the Introduction would help situate the proposed approach within the broader context."*

**Response:**

We appreciate your suggestion for expanding the background of the study. In fact, we briefly reviewed the main methods for urban flood water depth estimation in the original manuscript, including water level gauges, remote sensing technologies, and hydrodynamic models.

We pointed out that while water level gauges provide accurate point measurements, their high deployment and maintenance costs limit their widespread use. Remote sensing methods are mainly used to identify flood inundation areas, but they do not yet have the

capability to directly estimate water depth. While hydrodynamic models can estimate water depth, they require high-quality input data, are computationally complex, and have slower response times, making them less suitable for real-time emergency response. We believe that the background provided supports the validity of our method using social media images for flood water depth estimation and shows that it can serve as an effective complement to traditional methods. Due to space limitations, the discussion of these methods in the original manuscript was relatively brief, but it covers the core points.

**Changes in manuscript:**

We have expanded the Introduction (page 3, lines 62–69) to briefly mention other methods.

**6) On the Discussion Section:**

**Comment:**

*"The Discussion needs to be strengthened. I encourage the authors to include a critical evaluation of their method, its limitations, and a comparison with related studies—particularly those using alternative reference objects."*

**Response:**

Thank you for your valuable suggestion regarding the discussion section. We fully agree with your point that we need to more comprehensively assess the limitations of our method and compare it with existing research, particularly studies using other reference objects.

In the revised manuscript, we will modify the discussion section in accordance with your suggestion, providing a more detailed analysis of the method's limitations. We will also add comparisons with other studies to help readers better understand the relative strengths and weaknesses of our method.

**Changes in manuscript:**

We have expanded Discussion with limitations and comparative analysis on page25-27, lines 491-538.

**7) Minor comment:**

**Comment:**

*"Line 329: Please rephrase the sentence "The numbers on the image represent...".*
*The current wording is unclear and may confuse readers."*

**Response:**

Thank you for pointing this out. We have revised this sentence for clarity.

**Changes in manuscript:**

We have rephrased the sentence for clarity (page 17, line 337).

We sincerely thank both referees again for their detailed and constructive feedback.

We believe that these revisions have substantially strengthened the manuscript.

---

## Author Response (AR2)

**Title:** *Automated urban flood level detection based on flooded bus dataset using YOLOv8*
**Manuscript Number:** *egusphere-2024-4053*
**Authors:** *Yanbin Qiu, Xudong Zhou, Jiaquan Wan, Tao Yang, Lvfei Zhang, Yuanzhuo Zhong, Leqi Shen, and Xinwu Ji*

Dear Referee,

We are truly grateful for your meticulous review and valuable suggestions on our manuscript. Based on your comments, we have revised the manuscript point-by-point and provided detailed explanations for the relevant issues. We have strived to enhance the rigor and clarity of the paper through these improvements and sincerely hope to gain your approval. The specific responses are as follows:

**Referee #1**

**1) Comment:**

*"Line 57: In addition, in Texas alone… The language makes it appear that the ratio of evacuation related fatalities would be larger if other states were included, which is likely untrue. Suggest changing the sentence to: For example, in Texas during the XXX flooding event, it was estimated that approximately 75% of flood-related fatalities occurred during evacuation efforts via local roadways, primarily due to the lack of awareness regarding inundation depth in the surrounding areas. (Alizadeh Kharazi and Behzadan, 2021)"*

**Response:**

Thank you for this helpful suggestion. We have revised the sentence.

**Changes in manuscript:**

We have rephrased the sentence for clarity (page 2, line 58).

**2) Comment:**

*"Line 154: The authors state that images with water depth exceeding 100 cm are rare, although Table 1 shows roughly equal distribution across the 4 levels. Can the authors clarify if this is because they intentionally constructed a fairly balanced set?"*

**Response:**

Thank you for raising this point. We did not intentionally construct a class-balanced dataset. Images with water depth  exceeding 100 cm are indeed fewer in our raw

collection; however, within those few images we collected, there is a relatively high number of bus instances, which makes the instance-level counts for this depth range comparable to the other levels and thus appear approximately balanced in Table 2. Our annotations were performed objectively without deliberate class balancing.

**Changes in manuscript:**

In the revised manuscript, we have added a clarification on page 6, lines 167–169.

**3)Comment:**

*"Line 227: What does it mean by the validation set expanded from 108 to 198 mages? If the authors are referencing their previous manuscript version, the readers would have no information about those previous versions, therefore only the final experiment setup should be described."*

**Response:**

Thank you for the helpful comment. We have revised the sentence to improve clarity.

**Changes in manuscript:**

In the revised manuscript, we have rephrased the sentence for clarity on page 10, line 238.

**4) Comment:**

*"Validation set: Can the authors describe the split of validation set across the 4 levels? It could be included within Table 1, or a similar table could be added."*

**Response:**

Thank you for the suggestion. We have added the level-wise distribution of the validation set. This information is now presented in Table 2 and referenced in the Methods section.

**Changes in manuscript:**

We added a sentence in the Methods section (page 10, line 239) describing the distribution of the validation set across the four levels.

**5) Comment:**

*"Section 2.2.3: I suggest the authors add a statement about the difficulty in collecting a larger dataset of complex scenes, reflecting their response to my previous comments. This would help address this question for future readers."*

**Response:**

We appreciate this suggestion. The relevant statement has been added to Section 2.2.3.

**Changes in manuscript:**

In the revised manuscript, we have added a relevant statement on page 10, lines 251–255.

**6) Comment:**

*"Line 247: The statement seems to convey that the authors are assessing the general performance of YOLOv8 compared to previous models, however, they are only assessing the performance for their current dataset. Suggest changing the statement to: Although the introduction states that YOLOv8 is the latest algorithm in the YOLO series and has been known to perform better than earlier versions on a general image dataset, comparative analysis with earlier versions was performed for this dataset to quantify performance differences."*

**Response:**

Thank you for this helpful suggestion. We have revised the sentence.

**Changes in manuscript:**

We have rephrased the sentence for clarity (page 11, lines 264-266).

**7) Comment:**

*"Line 273: For improved clarity, suggest adding a statement like the following prior to defining IOU: Additionally, the metric Intersection over Union (IOU) was also calculated."*

**Response:**

Thank you for this helpful suggestion. We have revised the sentence.

**Changes in manuscript:**

We have rephrased the sentence for clarity (page 12, line 290).

**8) Comment:**

*"Line 342: The statement is unclear without a qualifier for what improved Level 3 and 4 performance. Suggest changing the statement to: Notably, the detection results for higher-risk categories (Level 3 and Level 4) show improved AP values in all models trained with augmented images, as evidenced by the Precision-Recall curves shifting closer to the upper-right corner."*

**Response:**

Thank you for this helpful suggestion. We have revised the sentence.

**Changes in manuscript:**

We have rephrased the sentence for clarity (page 15, lines 360-361).

**9) Comment:**

*"Line 461: The following statement does not appear to address the specific advantage of data augmentation, and can be removed since the next statement already addresses it: This improvement can be credited to the enhanced learning and generalization abilities of models."*

**Response:**

Thank you for this helpful suggestion. We have removed the sentence.

**Changes in manuscript:**

We have removed the sentence for clarity (page 24, line 479).

**10) Comment:**

*"Line 505: The clause - they are usually deployed only in limited areas - is unclear. What does deployment refer to? Suggest clarifying/removing the clause."*

**Response:**

Thank you for this helpful suggestion. We have removed the clause.

**Changes in manuscript:**

We have removed the clause for clarity (page 26, line 522).

**11) Comment:**

*"Line 517: Since edge-device deployment wasn't tested by the authors in this study, suggest providing a reference for this claim."*

**Response:**

Thank you for this helpful suggestion. We have provided a reference for this claim.

**Changes in manuscript:**

In the revised manuscript, we have provided a relevant reference. (page 26, line 533).

**12) Comment:**

*"Line 531: Suggest providing references for illumination-guided transformers and light-aware attention mechanisms."*

**Response:**

Thank you for this helpful suggestion. We have provided a reference for illumination-guided transformers and light-aware attention mechanisms.

**Changes in manuscript:**

In the revised manuscript, we have provided a relevant reference. (page 27, line 548).

**Referee #2**

**1) Comment:**

*"The details of data collection in Section 2.1.1 are insufficient: The article mentions that images are from "Baidu, Google, Douyin, and WeChat" but fails to specify the specific search keywords, time range, and screening criteria. These pieces of information are crucial for evaluating the representativeness and reproducibility of the data. It is necessary to supplement the spatiotemporal distribution statistics of data sources (such as the proportion of each platform, year distribution) and screening rules."*

**Response:**

We are grateful for your constructive suggestion. In the revised manuscript (Section 2.1.1), we have supplemented the specific time range for data acquisition, stating that the dataset covers the most recent five years. In addition, we have explicitly described the screening criteria, which ensures that buses remain valid, stable reference objects. These additions improve the clarity, reproducibility, and representativeness of our dataset.

**Changes in manuscript:**

In the revised manuscript, we have supplemented relevant explanations on lines 144–156.

**2) Comment:**

*"In Section 2.2.2, the hyperparameter settings use default hyperparameters (e.g., batch size = 16, epochs = 100) without explaining the basis. It is recommended to supplement the reasons for choosing the default values."*

**Response:**

Thank you for your comment regarding our use of default hyperparameters. We adopted the official default settings of YOLOv8 because these values were established by the Ultralytics team through large-scale pretraining and benchmarking on COCO ((Common Objects in Context)) and other datasets, and are recommended as a configuration that achieves a robust balance among detection accuracy, convergence stability, and computational efficiency. Designed for general downstream object-detection tasks, this configuration has been widely used in recent detection and recognition studies; we have added supporting references in the revised manuscript. In line with this common

practice—and considering our hardware resources and the need for stable training—we therefore used the official defaults.

**Changes in manuscript:**

In the revised manuscript, we have supplemented relevant explanations on page 10, lines 241–244.

**3) Comment:**

*"Some research work can be useful. Water inrush mechanism and variable mass seepage of karst collapse columns based on a nonlinear coupling mechanical model. Multiphysics modeling of thermal-fluid-solid interactions in coalbed methane reservoirs: Simulations and optimization strategies. Diffusion evolution rules of grouting slurry in mining-induced cracks in overlying strata."*

**Response:**

Thank you for providing the three related studies. We have added citations to these works in the Introduction of the revised manuscript.

**Changes in manuscript:**

Citations to the relevant literature have been added in the Introduction of the revised manuscript (lines 7 and 53).

**4) Comment:**

*"Section 3.1.1 mentions that mAP50-95 has increased by 20%, but there is no analysis of the performance of specific categories under different risk levels, making the significance of the indicator improvement unclear. It is suggested to decompose the mAP improvement ratio by risk level."*

**Response:**

Thank you for your valuable suggestion. In response, we have added a detailed analysis of the mAP50-95 performance across different flood risk levels in Section 3.1.1, as shown in the newly added Table 5.

**Changes in manuscript:**

A new table (Table 5) and corresponding analysis were added to Section 3.1.1 to present mAP50–95 improvements by flood risk level. (page 18, lines 389–401).

We sincerely thank both referees again for their detailed and constructive feedback. We believe that these revisions have substantially strengthened the manuscript.

---

## Author Response (AR3)

**Title:** *Automated urban flood level detection based on flooded bus dataset using YOLOv8*
**Manuscript Number:** *egusphere-2024-4053*

Dear Editor,

Thank you for your valuable feedback and for accepting our manuscript for publication. We sincerely appreciate the time and effort you have dedicated to reviewing our submission.

We have carefully considered the points you raised and have made the following adjustments:

1. We have thoroughly reviewed all references and ensured that they are correctly formatted and consistent throughout the manuscript. Additionally, we have addressed the specific issue you pointed out regarding the improper citation format in the text.

2. We have ensured that the same font is used consistently across all figures in the manuscript.

We hope these revisions align with the journal's formatting requirements. Please let us know if any further adjustments are necessary.

Thank you once again for your constructive suggestions.

Sincerely,
Yanbin Qiu